

# The surface aerosol optical properties in urban areas of Nanjing,
# west Yangtze River Delta of China
B. L. Zhuang[1, 3, *], T. J. Wang[1, 3, **], J. Liu[1, 2, 3], S. Li[1, 3], M. Xie[1, 3], Y. Han[1, 3], P. L.
Chen[1], Q. D. Hu[1], X. Q. Yang[1, 3], C. B. Fu[1, 3], J. L. Zhu[4]
[1] School of Atmospheric Sciences, Nanjing University, Xianlin Ave. 163, Nanjing 210023, China
[2] Department of Geography and Planning, University of Toronto, Toronto, M5S 3G3, Canada
[3] Collaborative Innovation Center of Climate Change, Jiangsu Province, China
[4] Department of Energy and Environment, Zhejiang Prov. Development Planning & Research Institute, Hangzhou
310012, China
[*] Corresponding author, E-mail: blzhuang@nju.edu.cn; Tel.: +862589681156; fax: +862589683797
[**] Corresponding author, E-mail: tjwang@nju.edu.cn; Tel.: +862589683797; fax: +862589683797
**Abstract:** Observational studies of aerosol optical properties are useful to reducing uncertainties
in estimating aerosol radiative forcing and forecasting visibility. In this study, the observed near-surface
aerosol optical properties in urban Nanjing are analyzed from Mar 2014 to Feb 2016. Results show that
near-surface urban aerosols in Nanjing are mainly from local emissions and the regions around. They
have lower loadings but are more scattering than in most cities in China. The annual mean aerosol
extinction coefficient (EC), single scattering albedo (SSA) and asymmetry parameter (ASP) at 550 nm
are 381.96 Mm$^{-1}$, 0.9 and 0.57, respectively. The aerosol absorption coefficient (AAC) is about one
order of magnitude smaller than its scattering coefficient (SC). However, the absorbing aerosol has
larger Ångström exponent (AAE) value, 1.58 at 470/660 nm, about 0.2 larger than the scattering
aerosols' (SAE). All the aerosol optical properties followed a near unimodal pattern, the ranges around
their averages accounting for more than 60% of the total samplings. Additionally, they have substantial
seasonality and diurnal variations. High levels of SC and AAC all appear in winter due to higher





aerosol and trace gas emissions. AAE (ASP) is the smallest (largest) in summer because of high
relative humidity (RH) which also causes considerably larger SC and smaller SAE, although intensive
gas-to-particle transformation could produce a large number of finer scattering aerosols in this season.
Seasonality of EC is different from the columnar aerosol optical depth. Larger AACs appear at the rush
hours of the day while SC and Bsp only peak in the early morning. Aerosols are fresher at daytime than
at nighttime, leading to their larger AE and smaller ASP. Different temporal variations between AAC
and SC cause the aerosols more absorbing (smaller SSA) in autumn and around rush hours. ASP has a
good quasi-LogNormal growth trend with increasing SC when RH is below 60%. The correlation
between AAC and SC at the site is close but a little smaller than that in suburban Nanjing in spring.
Atmospheric visibility decreases exponentially with increasing EC or SC, more sharply in spring and
summer. It could be further deteriorated with increasing SSA and ASP.
**1 Introduction**

Atmospheric aerosols have substantial influences on human health, air quality and climate changes

and their loadings have significantly increased since the preindustrial times. Due to their ability of
scattering/absorbing solar radiation and acting as cloud condensation nuclei, atmospheric aerosols can
affect atmospheric radiation and dynamics, as well as the Earth's hydrologic cycle, leading to regional
or global climate changes (Forster et al., 2007). Light scattering aerosols have contributed to offsetting
the warming effect of $CO_2$ (Kiehl and Briegleb 1993) while light absorbing aerosols such as black
carbon (BC) could further enhance the global warming (Jacobson 2002), especially in the high aerosol
regions. Due to the warming effect of BC, the atmosphere would become more unstable, which might
result in the changes in the trend of precipitation in China over the past decades as suggested by Menon





et al. (2002). Furthermore, atmospheric aerosols can be a major component in haze pollution, altering
atmospheric visibility and being harmful to human health (Chameides and Bergin, 2002).

Observations and modeling studies have been conducted on aerosol optical properties and

radiative forcing, as well as its climate effects on regional and global scales in the past two decades
(e.g., Penner et al., 2001; Bellouin et al., 2003; Liao and Seinfeld, 2005; Yan et al., 2008; Wu et al.,
2012; Zhuang et al., 2013a; 2014a; Wang et al., 2015; Yu et al., 2016). Forster et al. (2007) summarized
that large uncertainties exist in estimating the aerosol radiative forcing, especially in climate models.
The simulated global mean direct radiative forcing ranged from +0.04 to -0.63 W m$^{-2}$ for total aerosols
and from +0.1 to +0.3 W m$^{-2}$ for BC. This would further lead to much larger uncertainties in the
estimations of the aerosol climate effects. In East Asia, the range of simulated BC direct raidative
forcing is much larger than the global one, varying from +0.32 to +0.81 W m$^2$ (Zhuang et al., 2013a).
The bias is mostly resulted from the uncertainties in the simulated aerosol optical properties (Holler et
al., 2003), which, in turn, are related to the aerosol loadings, profiles, compositions, mixing states and
the atmospheric humidity. The uncertainty could be substantially reduced in a model if the aerosol
optical properties are corrected based on the observations or if the observed properties are directly used
(Forster et al., 2007).

In the last three decades, China has experienced the rapidest economic growth among East Asia

and even the world. This leads to high emission of aerosols and trace gases (Zhang et al., 2009). The
anthropogenic aerosol emissions in East Asia were estimated to exceed 1/4 of the global emissions
(Streets et al., 2001), resulting in more diversified aerosol compositions, complex species and
heterogeneous spatial distributions in the region (Zhang et al., 2012), especially in megacities and
urban agglomerations (e.g., Beijing-Tianjin-Hebei (BTH), Yangtze River Delta (YRD) and Pearl River





Delta (PRD) regions). Uncertainties in the aerosol radiative forcing and corresponding climate effects
might be much larger than in the rest of the world. Therefore, it is necessary to characterize the aerosol
optical properties based on observations in China, as did many studies in recent years at urban sites and
in rural areas (e.g., Bergin et al., 2001; Xu et al., 2002; 2004; Zhang et al., 2004; Yan, 2006; Xia et al.,
2007; Li et al., 2007; Yan et al., 2008; Andreae et al., 2008; He et al., 2009; Wu et al., 2009; Wang et al.,
2009; Li et al., 2010; Fan et al., 2010; Bai et al., 2011; Cai et al., 2011; Xiao et al., 2011; Xu et al.,
2012; Wu et al., 2012; Zhuang et al., 2015; Zhang et al., 2015; Li et al., 2015a; b; Yu et al., 2016). In
urban areas, Bergin et al. (2001) reported that the monthly mean aerosol scattering coefficient (SC at
530 nm) and absorption coefficient (AAC at 565 nm) were 488 and 83 $Mm^{-1}$, respectively, near the
surface in Beijing in June 1996. The annual mean 532 nm-AAC in Beijing was about 56 $Mm^{-1}$ from
2005 to 2006 (He et al. 2009) and it was 41~44 $Mm^{-1}$ in an urban site of YRD from 2012 to 2013
(Zhuang et al., 2015). Observations from Wu et al. (2009), Cao et al. (2012) and Tao et al. (2014)
suggested that the annual averaged aerosol optical properties were much larger in center to southwest
China and in PRD. The annual mean 520 nm-SC and 532 nm-AAC were 525 and 83 $Mm^{-1}$,
respectively, in Xi'an in 2009 and were 456 and 96 $Mm^{-1}$, respectively, in Chengdu in 2011. AAC was
about 82 ± 23 $Mm^{-1}$ in PRD. In rural and other areas, Xu et al. (2002; 2004) showed that 530 nm-SC
and 565 nm-AAC were 353 and 23 $Mm^{-1}$, respectively, at a rural site in YRD in Nov 1999, and were
158 and 6 $Mm^{-1}$, respectively in desert region (Yulin) in Apr 2001. Yan et al. (2008) reported that the
annual mean 532 nm-AAC and 525 nm-SC from 2003 to 2005 were 17.5 and 174.6 $Mm^{-1}$, respectively,
at a rural site in Beijing. In addition to surface measurements, the columnar optical properties of the
aerosols were also observed. Xia et al. (2007) indicated that the annual mean aerosol optical depth
(AOD) at 500 nm and its Ångström exponent (AE) in YRD were about 0.77 and 1.17, respectively.





Zhuang et al. (2014a) suggested that the AOD and AE of absorbing aerosols from 2011 to 2012 were
0.04±0.02 and 1.44±0.50, respectively, in urban Nanjing. Che et al. (2015) reported long-term
measurements of the countrywide-aerosol optical depths and Ångström exponents in China from 2002
to 2013. In spite of substantial observation-based studies mentioned, measurements and analysis on
aerosol properties in YRD region, one of the most populous regions in China, is still rather limited. To
fill the gaps in the current observational network in China and to better understand the optical
properties of urban aerosols in YRD, this study will analyze the observations of aerosol scattering (SC),
back scattering (Bsp), absorption (AAC), extinction (EC) coefficients and single scattering albedo
(SSA), Ångström exponent of scattering (SAE) and absorbing (AAE) aerosols, as well as aerosol
asymmetry parameter (ASP) in urban area of Nanjing, a major megacity in YRD. Our ultimate goals
are to reduce uncertainties in estimating aerosol radiative forcing and climate effect and to improve
forecast accuracy of visibility.
In the following, the method is described in Section 2. Results and discussions are presented in
Section 3, followed by Conclusions in Section 4.

**2 Methodologies**
**2.1 Sampling station and instruments**
The sampling station is located at the Gulou campus of Nanjing University, urban area of Nanjing
(32.05º N, 118.78º E). It is built on the roof of a 79.3 m-tall building, around which there are no
industrial pollution sources within a 30-km radius but there are several main roads with apparent traffic
pollution, especially at rush hours. The layout of the site and the corresponding climatology have been
described in Zhu et al. (2012).



The wavelength dependent aerosol absorption coefficient (AAC) and concentrations of black
carbon (BC) were derived from the measurements using a seven-channel Aethalomter (model AE-31,
Magee Scientific, USA). The wavelength dependent aerosol scattering coefficient (SC) and back
scattering coefficient (Bsp) were measured by a three-wavelength integrating Nephelometer (Aurora
3000, Australia). To make a brief comparison, the wavelength dependent columnar aerosol optical
depth (AOD) was observed using a Cimel sunphotometer (CE-318). The AE-31 model measures light
attenuation at seven wavelengths, including 370, 470, 520, 590, 660, 880, and 950 nm, respectively,
with a desired flow rate of 5.0 L/min and a sampling interval of 5 min. Aurora 3000 measures the
aerosol's light scattering, including SC and Bsp at 450, 525 and 635 nm, with a sampling interval of 1
min. CE-318 measures the AOD from 340 to 1640 nm at day times. Routine calibrations and
maintenances were carried out for all these instruments during the sampling periods. R-134 was used as
a span gas for Aurora 3000. The aerosol inlet is located about 1 m above the roof. Data to be analyzed
in this study were measured from Mar 2014 to Feb 2016 for AE-31 and CE-318 and from Jun 2014 to
Feb 2016 for Aurora 3000. Meteorological data during the sampling period are from the National
Meteorological Station of Nanjing (No. 58238).

**2.2 Calculation of the aerosol optical properties**
The wavelength dependent aerosol absorption coefficient (AAC) and BC mass concentration
can be calculated directly based on the measured light attenuations through a quartz filter matrix
(Petzold et al., 1997; Weingartner et al., 2003; Arnott et al., 2005; Schmid et al., 2006):
$$\sigma_{\mathrm{ATN},t}(\lambda) = \frac{(\mathrm{ATN}_t(\lambda) - \mathrm{ATN}_{t-1}(\lambda))}{\Delta t} \times \frac{A}{V} \qquad (1)$$
where $A$ (in m$^2$) is the area of the aerosol-laden filter spot, $V$ is the volumetric sampling flow rate (in





L/min) and $\Delta t$ is the time interval (=5 min) between $t$ and $t$-1. $\sigma_{\text{ATN}}$ is the AAC without any
correction, which is generally larger than the actual one ($\sigma_{\text{abs}}$) because of the optical interactions of
the filter substrate with the deposited aerosol. Generally, there are two key factors leading to the bias: 1)
multiple scattering of light at the filter fibers (multiple scattering effect), and 2) instrumental response
with increased particle loading on the filter (shadowing effect). Thus, the correction is needed and the
calibration factors $C$ and $R$ (shown in Eq. 2) are introduced to against the scattering effect and
shadowing effect, respectively:
$$\sigma_{\text{abs},t}(\lambda) = \frac{\sigma_{\text{ATN},t}(\lambda)}{C \times R} \tag{2}$$
Collaud Coen et al. (2010) suggested that AAC corrected from Weingartner et al. (2003) (WC2003 for
short, hereinafter) and Schmid et al. (2006) (SC2006 for short, hereinafter) have good agreements with
the one measured by a Multi-Angle Absorption Photometer. These two corrections are similar to each
other and they use the same $R(\lambda)$ :
$$R_t(\lambda) = (\frac{1}{f} - 1) \times \frac{\ln(\text{ATN}_t(\lambda)) - \ln 10}{\ln 50 - \ln 10} + 1 \tag{3}$$
where $R$ =1 when $\text{ATN} \leq 10$ and $f$ =1.2. However, $C$ value is fixed in WC2003 while is
wavelength dependent in SC2006. According to Wu et al. (2013) and Zhuang et al. (2015), $C$ in
Nanjing is 3.48 in WC2003 while it is 2.95, 3.37, 3.56, 3.79, 3.99, 4.51 and 4.64 at 370, 470, 520, 590,
660, 880, and 950 nm, respectively, in SC2006. Zhuang et al. (2015) further suggested that wavelength
dependent AACs corrected by SC2006 might be more close to the real ones than WC2003's in Nanjing,
although 532 nm-AACs from these two corrections are close to each other. In addition to the direct way,
AAC can also be calculated indirectly:
$$\sigma_{\text{abs},t}(\lambda) = [BC] \times \gamma \tag{4}$$





where [BC] is the mass concentration of Aethalometer BC (in μg/m$^3$) without any correction and $\gamma$ is
the conversion factor determined empirically from linear regression of the Aethalometer BC
concentration versus the aerosol absorption measurement (Yan et al., 2008). Zhuang et al. (2015)
indicated that $\gamma$ from the linear regression of the Aethalometer BC concentrations (ng/m$^3$) at 880 nm
against the light absorption coefficient (Mm$^{-1}$) at 532 nm in Nanjing is about 11.05 m$^2$/g. It's obviously
that only 532 nm-AAC can be addressed from this way. Thus, AACs corrected from SC2006 are used
in this study.

Based on wavelength dependent AAC and SC, Ångström exponent of scattering (SAE) and

absorbing (AAE) aerosols are estimated as followed:
$$AAE_{470/660\text{nm}} = -\log(AAC_{470nm} / AAC_{660nm}) / \log(470 / 660) \tag{5}$$
$$SAE_{450/635\text{nm}} = -\log(SC_{450nm} / SC_{635nm}) / \log(450 / 635) \tag{6}$$

For purposes of comparison, AAC at 450, 525, 532, 550 and 635 nm, SC at 532 and 550 nm

as well as Bsp at 532 and 550 nm were further calculated by the given coefficients and
corresponding Ångström exponents:
$$\sigma_\lambda = \sigma_{\lambda_0} \times (\frac{\lambda}{\lambda_0})^{-\alpha} \tag{7}$$
where, $\sigma_\lambda$ is the coefficient at wave length $\lambda$, $\alpha$ is the corresponding Ångström exponents.

Based on wavelength dependent SC, Bsp, AAC, aerosol asymmetry parameter (ASP), single

scattering albedo (SSA) and extinction coefficient (EC) are further estimated:
$$ASP_\lambda = -7.143889\beta_\lambda^3 + 7.46443\beta_\lambda^2 - 3.9356\beta_\lambda + 0.9893 \tag{8}$$
$$SSA_\lambda = \frac{SC_\lambda}{SC_\lambda + AAC_\lambda} \tag{9}$$
$$EC_\lambda = SC_\lambda + AAC_\lambda \tag{10}$$


where, $\beta_\lambda$ is the ratio of Bsp to SC at wavelength $\lambda$. Eq. 8 derives from Andrews et al. (2006).

**3 Results and discussions**

It's well known that the temporal variations of the aerosol optical properties at different

wavelengths are generally consistent with each other. Therefore, only single wavelength (such as 550
nm) AAC, SC, Bsp, SSA and ASP are focused when analyzing their basic characteristics (including
temporal variations, frequency distributions and changes with wind direction), their relationships with
each other, and their relationships with the meteorological conditions (such as RH and VIS) and
columnar AOD.
**3.1 Temporal variations of the aerosol optical properties**

The aerosol absorption coefficient (AAC) was directly obtained from the measurement of AE-31

and the scattering and back scattering coefficients (SC and Bsp) were directly measured from Aurora
3000. Based on wavelength dependent AAC and SC, Ångström exponent of absorbing (AAE at
470/660 nm) and scattering (SAE at 450/635 nm) aerosols were estimated according Eq.5 and Eq. 6,
respectively. Based on AAC, SC and Bsp, wavelength dependent aerosol asymmetry parameter (ASP),
single scattering albedo (SSA) and extinction coefficient (EC) are further estimated using Eq. 8~10 and
analyzed. Table 1 lists the statistical summary of surface aerosol optical properties in urban area of
Nanjing during the sampling period. The annual mean AAC, SC, Bsp, EC, SSA and ASP at 550 nm,
AAE at 470/660 nm and SAE at 450/635 nm is 29.615 Mm$^{-1}$, 338.275 Mm$^{-1}$, 44.257 Mm$^{-1}$, 381.958
Mm$^{-1}$, 0.901, 0.571, 1.583 and 1.320, respectively, with a standard deviation of 20.454 Mm$^{-1}$, 228.078
Mm$^{-1}$, 27.396 Mm$^{-1}$, 252.271 Mm$^{-1}$, 0.049, 0.088, 0.228 and 0.407, respectively.
**Table 1**


Figure 1 shows the 10th, 25th, median, 75th and 90th percentile values of the 550 nm- AAC, SC,
Bsp, 470/660 nm-AAE and 450/635 nm-SAE in urban area of Nanjing in each season from Mar 2014
to Feb 2016. Default values of the scattering aerosols' optical properties in spring 2014 are blank
because the measurements of Aurora 3000 started from June 2014. The figure suggests that AAC, SC,
Bsp, AAE and SAE have substantially seasonal variations. High level of AAC appears in winter times
(DJF) while the lower one is found in summer (JJA) (Fig. 1a). The temporal trend of Bsp is similar to
AAC's (Fig. 1d). According to Zhang et al. (2009), emissions of the aerosols and trace gases in China
are larger in winter than in the other seasons especially for carbonaceous aerosols (Fig. 1c in Zhuang et
al., 2013b). Thus, the higher AAC values in winter than in summer might be mostly resulted from the
higher aerosol emissions, lower boundary height and less rainfall. However, possibly due to the
impacts of RH in summer and dust aerosol in spring (Zhuang et al., 2014a), SC is considerably large in
these two seasons (Fig. 1c). Thus, the lowest SC is found in autumn in both 2014 and 2015. AAE has
seasonality similar to AAC. Due to RH, small value of AAE is found in JJA while the larger ones
appear in the other seasons (Fig. 1b), which is different from the seasonality of SAE. SAE is larger in
warmer seasons but is smaller in the other seasons. Scattering aerosols, including inorganic and
partially organic components, mainly come from gas-to-particle transformation, so that they have
smaller sizes (larger AE) compared to the primary aerosols (such as dust and BC). The efficiency of
gas-to-particle transformation is higher in warmer seasons. The observations of the aerosol
compositions at the site showed that seasonal mean inorganic aerosols, including sulfate, nitrate and
ammonium, account for about 50% of the total $PM_{2.5}$ in spring and might be higher than 50% in the
other seasons (Zhuang et al., 2014b). Thus, SAE in summer and autumn is large (Fig. 1e). RH can
impose substantial influences on scattering aerosols. SAE might be much larger than the current values





in these two seasons if the moisture absorption growing were excluded. Seasonal mean RH is about
75.41% and 70.86% in JJA and SON, respectively, to a certain degree leading to higher values of SAE
in autumn than in summer. The figure also suggests that aerosol absorption coefficient and scattering
coefficient as well as their sizes in 2014 are higher than those in 2015, which might somewhat relate to
a difference in RH in these two years. A comparison of RH between 2014 and 2015 indicates that RH is
79.49% and 72.86% in JJA and SON, respectively, in 2014, larger than that in 2015 (71.33% in JJA and
69.03 in SON).
**Figure 1**
Figure 2 plots the seasonal mean values with standard deviations of AAC, SC, Bsp, EC, SSA, ASP
at four wavelengths, AAE at 470/660 nm and SAE at 450/635 nm. AAC, SC, Bsp and EC increase with
decreasing wavelength in four seasons. Changes in SSA and ASP with increasing wavelength are
different in different seasons. SSA increases with increasing wavelength in colder seasons but little in
JJA and SON. ASP increases with wavelength in JJA, opposite to in other seasons. The figure also
suggests that seasonal variation of EC is more consistent with SC's, with large values in JJA and DJF
(370.236 and 422.569 Mm$^{-1}$, respectively, at 550 nm). The largest values of SSA and ASP appear in JJA
(0.933 and 0.638, respectively, at 550 nm), implying that aerosols in urban area of Nanjing are more
scattering and have stronger forward scattering ability in JJA than in other seasons. The urban aerosols
are more absorbent in SON in Nanjing (550 nm SSA is about 0.874).
**Figure 2**
Seasonal mean 550 nm AAC, SC, Bsp, EC, SSA, and ASP, 470/660 nm AAE and 450/635 nm
SAE as well as corresponding standard deviations are listed in Table 2. It suggests that seasonal mean
550 nm AAC, SC, Bsp, EC, SSA, and ASP vary from 19.65 to 37.96 Mm$^{-1}$, 294.62 to 385.14 Mm$^{-1}$,





36.99 to 54.79 Mm$^{-1}$, 341.3 to 422.57 Mm$^{-1}$, 0.874 to 0.933, and 0.54 to 0.64, respectively. Seasonal
mean AAE and SAE vary from 1.49 to 1.70 and 1.1 to 1.54, respectively. AAC and Bsp in DJF are
about 2 and 1.5 times of those in JJA, respectively. SSA in JJA is about 6.75% larger than that in SON.
**Table 2**

In addition to seasonality, the aerosol optical properties near the surface at urban Nanjing have

substantial diurnal variations (Figure 3), especially for the coefficients (AAC, SC, Bsp and EC). The
diurnal variation of EC, which is consistent with SC, is not showed in the figure. AAC levels are
usually high at the rush hours around 07:00-09:00 am and around 09:00-11:00 pm but low in the
afternoon (Fig. 3a). At 08:00 am, mean 550 nm-AAC is as large as about 34 Mm$^{-1}$, while at 02:00 pm,
it is about 23 Mm$^{-1}$. SC and Bsp (Fig. 3b and 3c), to some extent, have diurnal variations similar to
AAC's. Their lowest values also appear in the afternoon (about 280 Mm$^{-1}$ for SC and 38 Mm$^{-1}$ for Bsp).
However, only one peak of the aerosol scattering coefficient is found in the early morning (about 379
Mm$^{-1}$ for Sc and 48 Mm$^{-1}$ for Bsp) and it is about 1-2 hours earlier than its absorption coefficient
possibly owing to the different emissions between these two types of aerosols. Absorbing aerosols in
urban Nanjing mainly come from the vehicle emissions because of the developed transportation
network, resulting in two peaks of AAC within one day (Zhuang et al. 2015). Scattering aerosol
loadings are somewhat less affected by traffic emissions especially in nighttime. Their precursors, such
as SO$_2$ and NOx, are mostly come from coal combustion and industrial emissions in urban Nanjing
based on source apportionment. Therefore, there is no peak for SC or Bsp before midnight, although
their values are considerably large (about 350 and 46 Mm$^{-1}$, respectively). Different diurnal cycles
between AAC and SC were also observed in sub-urban area of Nanjing (Yu et al., 2016). Diurnal
variations of AAC, SC and Bsp might be highly affected by the diurnal cycles of the boundary layer.





The small coefficients in afternoon are mostly induced by well developed mixing layer (Zhuang et al.
2014b). Generally, the boundary layer becomes more and more stable after sunset and its height
becomes lower, which is conducive to the accumulation of air pollutants in the nighttime especially
during the period from midnight to sunrise. Therefore, SC usually peaks in early morning and the peak
appears at different times in different seasons (05:00 am in JJA and 08:00-09:00 am in DJF). The
daytime peak of AAC appears at 07:00 am in JJA and at 09:00 am in DJF. Diurnal variation of SSA
also reflects the difference between AAC and SC (Fig. 3d), implying that aerosols in urban Nanjing are
more scattering after midnight (SSA is about 0.91) while more absorbing before noon and midnight
(SSA is about 0.89). Additionally, SSA is also large in afternoon possibly because the dilution effect of
well developed boundary layer on scattering aerosol is weaker than that on absorbing aerosols.
Scattering aerosols mainly come from strong chemical production (gas-to-particle transformation) at
daytime, which to some extent offsets the dilution effect of the boundary on SC. The figure further
shows that both AAE (Fig. 3e) and SAE (Fig. 3f) at daytime are slightly larger than those after
midnight because both absorbing and scattering aerosols are more fresher at daytime while they are
more aged before sunrise. Diurnal variations of SAE and AAE are relatively weaker compared to
corresponding coefficients. In addition to aerosol loadings, the level of Bsp is also affected by the size
of the aerosols as suggested by Yu et al. (2016), so is ASP (Fig. 3g). Diurnal cycle of ASP is similar to
that of Bsp but is opposite to that of SAE. Large ASP appears in early morning (0.587) and the lower
ASP in afternoon (0.552).
**Figure. 3**
**3.2 Frequencies of the aerosol optical properties**
The frequency of the aerosol optical properties is presented in Figure 4. Similarly, the frequency



of EC is not shown in the figure because it has similar pattern to SC's. Almost all of them follow a
unimodal pattern. The dominant range is from 9 to 45 Mm$^{-1}$ for AAC, 60 to 390 Mm$^{-1}$ for SC, 15 to 60
Mm$^{-1}$ for Bsp, 0.87 to 0.97 for SSA 1.4 to 1.8 for AAE, 0.96 to 1.68 for SAE and 0.48 to 0.69 for ASP,
accounting for over 73%, 67%, 69%, 73%, 71%, 62% and 81% the total samplings during the entire
study period, respectively. The maximum frequencies of 32.9% (AAC), 24.04% (SC), 26.45% (Bsp),
18.64% (SSA), 20.9% (AAE), 18.06% (SAE) and 34% (ASP) occur in the ranges from 9 to 21 Mm$^{-1}$,
170 to 280 Mm$^{-1}$, 30 to 45 Mm$^{-1}$, 0.91 to 93, 1.5 to 1.6, 1.32 to 1.5 and 0.55 to 0.62, respectively.
Frequency distributions of the aerosol optical properties have substantially seasonal variations. The
frequency peaks of the properties would be more concentrated at lower/higher ranges if their seasonal
means are smaller/larger. As shown in Fig. 4a, 4c, and 4e, relatively larger values or the peaks of
frequencies for AAC, Bsp and AAE are concentrated in lower value ranges in JJA but in higher value
ranges in the other seasons. Moisture absorption growth of absorbing aerosols leads to a left-ward shift
in an AAE-frequency curve in JJA. Effects of dust aerosol also might result in a left-ward shift in a
SC-frequency curve in spring (Fig. 4f). Furthermore, due to dust and RH, SC is considerably large in
MAM and JJA, leading to relatively larger frequencies of SC distributed at larger SC ranges compared
with the ones of AAC. As mentioned above, aerosols in urban Nanjing are more scattering and have
stronger forward scattering ability in JJA than in the other seasons, thus larger frequencies occur more
at higher value ranges of SSA and ASP in JJA.
**Figure 4**
**3.3 Aerosol optical properties in different wind directions**

East Asian monsoon is active in middle latitudes. Nanjing could be affected by East Asian summer

monsoon in JJA and by the winter monsoon in DJF. Air flows in these two seasons are significantly



different (Figure 5a and 5b) so to alter the aerosol optical properties in different seasons. Air masses
mostly come from the oceans (about 77%) in JJA and from continental regions in north and northwest
of China (57%) in DJF. Only a few percentages of air masses are from the north region of China in JJA.
Additionally, considerable air masses arriving at the site are from the local areas (cluster 1 in JJA) or
from places near Nanjing (cluster 1 in DJF). Therefore, the aerosol optical properties at the study site
are characterized differently with different air masses in the two seasons.
As suggested by Zhuang et al. (2014b), high BC loadings in early June 2012 were observed at the
site when the air masses were from northwestern directions of Nanjing, in which seriously biomass
burning was detected. Therefore, the aerosol optical properties are further analyzed by their origins in
both JJA and DJF (Fig 5c and 5d). In JJA, seasonal mean AAC, SC, Bsp, SSA, ASP, AAE and SAE are
about 19.65 $Mm^{-1}$, 340.87 $Mm^{-1}$, 36.99 $Mm^{-1}$, 0.93, 0.64, 1.49 and 1.34, respectively. The dominant air
masses are from local areas (cluster 1 in Fig. a) and east ocean (on the way through urban
agglomeration regions (cluster 2) and less-developed regions (cluster 3) of the Yangtze River Delta
YRD), accounting for 90% of the total characteristics of the aerosol optical properties in urban Nanjing.
All the values of the properties in the first three clusters are more close to their season means. Aerosol
absorption and scattering coefficients from local emissions are larger than those in the other clusters.
Although air masses in cluster 2 and cluster 3 come from the oceans and have the same level of relative
humidity (RH), differences still exist between the clusters. The air masses have to cross the urban
agglomeration (from Shanghai to Nanjing) of YRD when they arrive Nanjing in cluster 2 but pass less
developed regions (north Jiangsu Province) in cluster 3. In YRD, emissions of the aerosols and trace
gases are much stronger in urban agglomeration regions than those in other area as suggested in Zhang
et al. (2009) and Zhuang et al. (2013b). Therefore, AAC and SC in cluster 2 are larger than those in



331 cluster 3 to some extent (Fig. 5a and 5c). Aerosols from these two clusters are more scattering than the

332 local ones. There are two clusters (cluster 4 and 5 in Fig. 5a) from the remote areas in JJA. Aerosol

333 loadings are relatively small when the air masses from these two clusters. The size of the aerosols is

334 finer (larger AAE in cluster 5 and SAE in cluster 4 and 5 in Fig. 5c). ASP varying with the clusters

335 coincides with RH varying with the clusters (Fig. 5c), implying that RH might influence ASP

336 significantly. In DJF, seasonal mean AAC, SC, Bsp, SSA, ASP, AAE and SAE are about 37.96 Mm$^{-1}$,

337 385.14 Mm$^{-1}$, 54.79 Mm$^{-1}$, 0.89, 0.54, 1.70 and 1.24, respectively. Similar to JJA, the aerosol

338 absorption and scattering coefficients are the largest, all of which (AAC, SC and Bsp) are about 1.3

339 times of their season means (Fig. 5d), when the air masses are local or from the regions (cluster 1 in

340 Fig. 5b) near Nanjing in DJF. AAC, SC, Bsp, SSA and ASP are small but AAE and SAE are large if air

341 masses are from remote areas. Aerosols are the smallest, most absorbing and finest when the air masses

342 are from near Lake Baikal. ASP varying with the clusters also coincides with RH varying with the

343 clusters in this season (Fig. 5d), further implying the effect of RH on ASP.

344 **Figure 5**

345 Substantial studies on the aerosol optical properties have been carried out in China from monthly

346 to annual scales. Table 3 lists some annual and seasonal statistics of measured surface aerosol optical

347 properties from literature. Annual and season means listed in the table are comparable to some extent,

348 although the observational periods and instruments are different. It suggests that AACs and SCs in

349 urban areas are much higher than those in rural and remote areas. In Beijing (center of

350 Beijing-Tianjin-Hebei region), annual mean AAC and SC were 56 and 288 Mm$^{-1}$ in urban site during

351 the period from 2005 to 2006 (He et al., 2009), which were much larger than the ones (17.5 and 174.6

352 Mm$^{-1}$, respectively) in rural area (Yan et al., 2008). In Chengdu (Tao et al., 2014), Xi'an (Cao et al.,




2012) and Wuhan (Gong et al., 2015), which is the center from southwest to central China, the annual
mean scattering coefficients in these cities exceeded 450, 520 and 370 Mm$^{-1}$, respectively. In Pearl
River Delta (PRD) region, seasonal mean AAC at 532 nm was about 84 and 188 Mm$^{-1}$ at an urban site
(Panyu), about 47 and 95 Mm$^{-1}$ at a suburban site (Dongguan), about 26 and 28 Mm$^{-1}$ at a rural site,
and only 7.21 and 8.37 Mm$^{-1}$ at a remote site (Yongxing Island), in spring and winter, respectively (Wu
et al., 2013). Additionally, aerosols in urban areas are more absorbing. The aerosol absorptions in urban
areas have stronger seasonality than those in rural areas (Table 3). Urban aerosols in Nanjing in annual
scale are somewhat lower but more scattering than those in most cities in China. In addition to annual
and seasonal means, there are considerable studies on monthly mean aerosol optical properties (e.g.,
Bergin et al., 2001; Xu et al., 2002; 2004; Li et al., 2007; Andreae et al., 2008; Li et al., 2015a; b). A
few studies on the aerosol optical properties in Nanjing have been carried out previously (Zhuang et al.,
2014a; 2015; Yu et al., 2016) based on observations. They were more focused on the columnar aerosols
(Zhuang et al., 2014a), or single optical property (Zhuang et al., 2015), or shorten observations (two
months in Yu et al., 2016). Substantial analysis in the key optical properties of the surface aerosol here
to a certain degree fill the gaps in the study on the aerosols in Nanjing, even in YRD.
**Table 3**
**3.4 Relationship among aerosol optical properties, relative humidity and visibility**
The relationships between SC and AAC, SC and Bsp are presented by season in Figure 6. As
shown in Figures 3 and 4, these three types of coefficients have similar diurnal and frequency
distributions. It is obviously that relations between SC and Bsp are much better than those between SC
and AAC in all seasons. The linear correlation coefficient varies from 0.93 to 0.97 for SC and Bsp and
from 0.66 to 0.87 for SC and AAC in urban Nanjing. The correlation between AAC and SC becomes





poorer in MAM (0.66) and JJA (0.87) because the scattering aerosols is more affected by dust in spring
and SC is more affected by RH in summer. The linear correlation coefficients between SC and AAC
and between SC and Bsp in MAM at the site were a little smaller than that in suburban Nanjing (Yu et
al., 2016) in the same season in 2011. The slope of the fitting between Bsp and SC represents the levels
of ASP. Analysis (not shown) suggests that ASP has a significant anti-correlation with the ratio of Bsp
to SC (linear R=-0.98). Thus, a greater slope of curve represents a smaller ASP, thus less forward
scattering of the aerosols.
**Figure 6**

The correlations between ASP and SC under different RH conditions are illustrated in Figure 7,

showing that ASP has a quasi-LogNormal distribution with SC especially in lower RH conditions. ASP
increases monotonically with increasing SC in low RH ranges (Fig. 7a and 7b, RH < 60%) and ASP
mostly concentrates at small SC regions when RH is less than 40% (Fig. 7a), implying that fine
particles dominates the most in low RH conditions as also suggested by Andrews et al. (2006) and
Badu et al. (2012). The correlation between ASP and SC becomes poorer with increasing RH (Fig. c),
indicating that both fine and coarse aerosols might be equally important to the total SC.
**Figure 7**

Figure 8 shows the relationships between the SSA at 491 nm and extinction Angstrom exponent

(EAE) at 491/863 nm (Fig. a) as well as between SSA difference (863 nm - 491 nm) (short for dSSA)
and EAE at 491/863 nm (Fig. b). Overall, SSA or dSSA to a certain degree have an anti-correlation
with EAE in urban area of Nanjing, especially for the latter one. Linear correlation coefficient is about
-0.13 between SSA and EAE and about -0.75 between dSSA and EAE. Relationships between the SSA
(or dSSA) and EAE to some extent reflect the aerosol types and sources as indicated by Russell et al.
(2014), who proposed a method to identify the aerosol types based on the columnar aerosol optical
properties (including SSA, EAE and the real refractive index) from the Aerosol Robotic Network
(AERONET) retrievals. They suggested that: 1. The polluted dust aerosol had smaller EAE (near 1.0)
and SSA ranged from 0.85 to 0.95. 2. The urban aerosols had larger EAE values (around 1.4) and SSA
ranges (0.86~1.0) compared with the dust aerosols. 3. The biomass burning aerosol (dark type) had the
largest EAE (exceeding 1.5) while smaller SSA (about 0.85). If there were two kind of aerosols having
nearly identical coordinates in SSA and EAE, further information (such as the real refractive index)
should be used (Russell et al., 2014). Based on this method, the figure further implies that, in addition
to local emissions, aerosols in urban area of Nanjing might also be affected substantially by the long
distance transported dust (or polluted dust) in spring and be influenced to some extent by biomass
burning in fall.
**Figure 8**
Atmospheric humidity has significant influences on the growth of particulate matter, subsequently
affecting the sizes and absorbing/scattering abilities of the aerosols. As shown in Figure 7a and 7c, high
levels of SC are likely found in high RH ranges. Seasonal mean RH is the largest in summer but lowest
in winter (Figure 9a). Due to the effects of RH in summer, the aerosol scattering efficiency would be
enhanced substantially (Fig. 1c). Additionally, the smallest AAE in JJA corresponds to the highest RH,
and vice versa (Fig. 1b), indirectly verifying the effects of RH on the size of absorbing aerosols, i.e.,
coarser in high RH but finer in low RH. These results are consistent with Zhuang et al. (2014a), in
which characteristic of columnar aerosol optical properties were investigated. Figure 9b further shows
that AAE and SAE decrease monotonically with increasing RH. The correlation between ASP and RH
is opposite to that between aerosol Ångström exponent and RH, implying that the forward scattering





efficiency increases with increasing in RH. The linear correlation coefficients are -0.36, -0.15 and 0.6
between AAE and RH, SAE and RH, and ASP and RH, respectively, in urban areas of Nanjing. The
relation between ASP and RH is the best among these three optical properties, which has somewhat
shown in Fig. 2f, Fig. 5c and Fig. 5d. These results could be used to correct the aerosol optical
parameters in numerical models for estimating the aerosol radiative forcing in East China as suggested
by Andrews et al. (2006), in hope to reduce uncertainties in such estimation.
**Figure 9**

High levels of aerosol loadings would directly affect the visibility (VIS), which is one of the

factors being concerned about in current air quality forecasting in China. The forecast accuracy of
visibility or haze pollutions would be increased significantly if the effects of aerosols on visibility can
be figured out. Instead of the loadings of the particulate matter, the aerosol optical properties here are
used when investigating the aerosol effects on VIS.

Figure 10 shows the relations between extinction coefficient (EC) and VIS and between SC and

VIS by season under different RH levels. Atmospheric VIS is found to decrease exponentially with
increasing EC or SC in all seasons. The lapse rate of VIS with EC or SC is much larger in spring and
summer than in fall and winter. The lower VIS always appears at higher RH ranges, and vice versa. In
small VIS regions (such as: <4 km), VIS values are much smaller in JJA than those in the other seasons
under the same SC level, implying the strong effects of RH on VIS. The effect of AAC on VIS has
substantial seasonality and it is strong in SON but weak in MAM and JJA as illustrated in the fitting
lines in the figure. Study on the effects of PM on VIS might be more reasonable if using the aerosol
optical properties rather than its mass concentrations. The linear correlation coefficient between EC and
VIS varies from -0.69 (in JJA) to -0.87 (in DJF), and between SC and VIS, it varies from -0.71 (in JJA)





to -0.87 (in DJF) in urban area of Nanjing.
**Figure 10**
In addition to the SC or EC, the aerosol SSA and ASP also have good relationships with VIS as
shown in Figure 11, in which the effects of RH and SAE are also included (larger markers represent
smaller SAE, but larger size of the aerosols). The aerosols become coarser, less absorbing and more
forward scattering with increasing RH, which subsequently further exacerbate the deterioration of
visibility in all the seasons. The linear correlation coefficients vary from -0.48 (in JJA) to -0.73 (in
SON) between SSA and VIS and -0.47 (in JJA) to -0.80 (in MAM) between ASP and VIS in urban
Nanjing. These results additionally illustrate that the scattering aerosols are still the key factors
affecting the atmospheric visibility, although the absorbing aerosols might have considerable influences
on VIS in some seasons (Fig. 10c). The results in this study further indicate that effects of aerosols on
air quality are complex.
**Figure 11**
Comparison between surface aerosol extinction coefficient and columnar AOD is performed
(Figure 12). Differences exist between EC and AOD, although they are well correlated with each other
in each season. AOD to some extent is less affected by the development of boundary layer and more
affected by the transport of aerosols compared to EC at the surface. The seasonal mean EC is large both
in JJA and in DJF while the largest AOD is only found in JJA, which is possibly related to higher
boundary layer height in JJA. A lower boundary layer would lead to more aerosol accumulation at the
surface thus result in its smaller column burden. These differences (high surface aerosol loadings but
low AOD) have also been simulated by a regional climate chemistry model in Zhuang et al. (2011 and
2013). Overall, high AOD level corresponds to large EC value in each season, implying that aerosols in



the upper layers mostly come from surface emissions in urban Nanjing. In some cases, long distance
transport of aerosols might contributes significantly to the AOD as shown in Fig. 12a, in which AOD
exceeds 2 meanwhile EC is found to appear in low value ranges. The slope of the linear fitting is larger
in JJA (about 0.0016) than that in the other seasons (all about 0.001), indicating that for a given value
of EC, AOD would be higher in JJA possibly because of higher humidity in summer. The columnar
water vapor in summer is about 2 to 5 times of that in the other seasons.
**Figure 12**
**4 Conclusions**

In this study, the near-surface aerosol optical properties, including aerosol scattering (SC), back

scattering (Bsp), absorption (AAC) and extinction (EC) coefficients, single scattering albedo (SSA),
scattering (SAE) and absorbing (AAE) Ångström exponent, as well as asymmetry parameter (ASP), are
investigated based on the measurements with the 7-channel Aethalometer (model AE-31, Magee
Scientific, USA) and three-wavelength integrating Nephelometer (Aurora 3000, Australia) in urban
area of Nanjing from Mar 2014 to Feb 2016.

In urban area of Nanjing, the annual mean EC, SSA and ASP at 550 nm are 381.958 Mm$^{-1}$, 0.901,

0.571, respectively. SC, which accounts for about 90% of EC, is about one order of magnitude larger
than AAC, implying that EC to a great degree has similar temporal variation and frequency distribution
to SC. Absorbing aerosol is finer than the scattering one. AAE at 470/660 nm is about 1.58, about 0.2
larger than SAE. All of them above have substantially seasonal and diurnal variations. Both the aerosol
absorption and scattering coefficients have the largest values in winter due to the higher emissions.
However, SC also has a higher values in summer and spring likely due to higher relative humidity (RH)
and efficiency of gas-to-particle transformation in summer and the effect of dust in spring, respectively.





High RH in summer results in the lowest AAE and largest ASP being found and it is also lead to a
relatively smaller SAE, although a large number of fine scattering aerosols could be produced through
intensive gas-to-particle transformation in this season. Seasonality of SSA is co-determined by AAC
and SC, showing the largest value in summer and lowest value in fall. AAC, SC, Bsp and EC have
more substantial diurnal variations than SSA, AAE, SAE and ASP. Because of traffic emissions, AACs
are high at the rush hours (around 09:00 am and pm) but low in afternoon when the boundary layer
being well developed. SC and Bsp usually peak in the early morning before sunrise (1-2 earlier than
AAC's) and reach the bottom in the afternoon. High levels of SC and Bsp are mostly caused by
accumulation of air pollution in the nighttime from midnight to sunrise. The diurnal variation of SSA is
also depended on AAC and SC. SSA is large after midnight and noon. AAE and SAE at daytime are
slightly larger than after midnight because both absorbing and scattering aerosols are fresher at daytime
but more aged before sunrise. ASP, which is related to the size of the aerosols, its diurnal variation is
opposite to SAE's but similar to Bsp's.
Frequency analysis indicates that almost all of the aerosol optical properties follow a unimodal
pattern in urban area of Nanjing. The dominant ranges are from 9 to 45 Mm$^{-1}$ for AAC, 60 to 390 Mm$^{-1}$
for SC, 15 to 60 Mm$^{-1}$ for Bsp, 0.87 to 0.97 for SSA 1.4 to 1.8 for AAE, 0.96 to 1.68 for SAE and 0.48
to 0.69 for ASP, accounting for more than 73%, 67%, 69%, 73%, 71%, 62% and 81%, respectively, of
the total data samples during the entire study period. Frequency distributions of the aerosol optical
properties also have substantial seasonality. The frequency peak of a property would be more
concentrated among lower/higher ranges if the seasonal mean is smaller/larger. Back trajectory analysis
suggests that the source of aerosols in Nanjing are mainly from the local and regional emissions around
YRD in summer, while from the sources include both local emissions and transport from central and



north China in winter. In JJA, aerosols are more scattering when air masses come from the East China
Sea and finer if air masses come from remote areas. In DJF, AAC, SC, Bsp, SSA and ASP are low
while AAE and SAE are high in urban Nanjing under the conditions of air masses being transported
from remote areas. ASP varied with the clusters is consistent with RH in both JJA and DJF.

The correlation between SC and Bsp is much better than that between SC and AAC in all seasons.

In spring, these relationships are a little weaker than those in suburban Nanjing. ASP has a
quasi-LogNormal distribution with SC under a condition of RH being lower than 60%, increasing
monotonically with increasing SC. It would be mostly concentrated at small SC regions when RH is
less than 40% because finer particles dominant under low RH conditions. The correlation between ASP
and SC becomes weaker with increasing RH, indicating that both fine and coarse aerosols might be
equally important to the total SC in high RH conditions. Atmospheric humidity can significantly
modulate aerosol optical properties. Due to the effects of RH in summer, the aerosol would become
coarser and its forward scattering efficiency would be stronger with increasing in RH. The linear
correlation coefficients are -0.36, -0.15 and 0.6 between AAE and RH, SAE and RH, and ASP and RH,
respectively, in urban areas of Nanjing. Comparisons also indicate that seasonal variation of surface
aerosol EC (high in JJA and DJF) is different from its columnar optical depth (AOD, high in JJA and
low in DJF), even though they are closely correlated to each other within each season. Overall, high
AOD level corresponding to large EC value in each season implies that aerosols in upper layers are
mostly from surface emissions. AOD would be higher in JJA than in other seasons in a condition with
fixed EC, possibly due to the effects of high humidity.

Overall, the scattering aerosols are still the key factor in affecting the atmospheric visibility

(VIS), although the absorbing aerosol has considerable contributions in some seasons. The linear



correlation coefficient between EC and VIS varies from -0.69 to -0.87, close to those between SC and
VIS. VIS is found to be decreased exponentially with increasing EC or SC in all seasons. And its lapse
rate along with EC or SC is much larger in spring and summer than in fall and winter. In small VIS
regions (i.e., VIS<4 km), VIS values are much smaller in JJA than in other seasons if the SC levels are
the same, further indicating the strong effect of RH on VIS. The aerosol SSA and ASP could also affect
VIS. Large SSA and ASP might further exacerbate the deterioration of visibility. The linear correlation
coefficients between seasonal SSA and VIS varies from -0.48 to -0.73 and from -0.47 to -0.80 between
ASP and VIS in urban area of Nanjing.

**Acknowledgements:** This work was supported by the National Key Basic Research Development
Program of China (2014CB441203), the National Natural Science Foundation of China (91544230,
41475122), the New Teachers' Fund for Postdoctoral Fellows, Ministry of Education
(20120091120031), FP7 project: REQUA (PIRSES-GA-2013-612671), and a project Funded by the
Priority Academic Program Development of the Jiangsu Higher Education Institutions (PAPD). the
National Science Foundation of Jiangsu Provence (Grant #BE2015151). The authors would like to
thank all members in the AERC of Nanjing University for maintaining instruments. The HYSPLIT
model was supplied by NOAA: http://ready.arl.noaa.gov/HYSPLIT_traj.php.





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



**Figure captions:**


Figure 1. The 10th, 25th, median, 75th, and 90th percentiles of 550 nm AAC (a, Mm⁻¹), 470/660 nm


AAE (b), 550 nm SC (c, Mm⁻¹), 550 nm (d, Mm⁻¹) and 450/635 nm SAE (e) in each season from March


2014 to Feburay 2016.


Figure 2. Seasonal means (markers) and corresponding standard deviations (error bars) of wavelength


dependent AAC (a, Mm⁻¹), SC (b, solid mark, Mm⁻¹), Bsp (b, open mark, Mm⁻¹), EC (c, Mm⁻¹), SSA (e)


and ASP (f) at 450, 532, 550, 635 nm, as well as AAE at 470/660 nm (d, red solid mark) and SAE at


450/635 nm (d, green open mark)


Figure 3. Diurnal variations of 550 nm AAC (a, Mm⁻¹), SC (b, Mm⁻¹), Bsp (c, Mm⁻¹), SSA (d), ASP (g),


470/660 nm AAE (e) and 450/635 nm SAE (f) during the study period.


Figure 4. Frequency (%) distributions of 550 nm AAC (a), SC (b), Bsp (c), SSA (d), ASP (g), 470/660


nm AAE (e) and 450/635 nm SAE (f) on annual (shaded bar) and seasonal (markers in colors) scales.


Figure 5. Clusters of 96-h back trajectories arriving at the study site at 100 m in JJA (a) and DJF (b)


simulated by the HYSPIT model. The means with standard deviations of the aerosol optical properties


at each cluster of back trajectories in both JJA and DJF are plotted in Fig. 5c and 5d, respectively.


Figure 6. Relationships between 550 nm AAC and SC (solids square in blue) and between 550 nm Bsp


and SC (solid cycles in gray) in spring (a), summer (b), autumn (c) and winter (d).


Figure 7. Relationships between the 550 nm ASP and SC in different RH levels.


Figure 8. Relationships between the 491 nm SSA and extinction Angstrom exponent (EAE) at 491/863


nm (a) and between the SSA difference (863-491 nm) and EAE at 491/863 nm (b).


Figure 9. Seasonal variations of RH (a, %) and linear correlations between AAE and RH (b, light blue,


upper), between SAE and RH (b, green, middle), and between ASP and RH (c, deep blue, lower).






Figure 10. Relationships between SC and visibility (open cycles) and between EC and visibility (solid
cycles) in different RH levels in spring (a), summer (b), autumn (c) and winter (d).
Figure 11. Relationships between SSA and visibility (solid cycles) and between ASP and visibility
(solid squares) in different RH and AAE levels in spring (a), summer (b), autumn (c) and winter (d).
Figure 12. Relationships between surface EC at 550 nm and column AOD at 500 nm in spring (a),
summer (b), autumn (c) and winter (d).

**Table captions:**
Table 1 Statistical summary of the surface aerosol optical properties in Nanjing.
Table 2 Seasonal mean±SD of the surface aerosol optical properties in Nanjing.
Table 3 The aerosol optical properties in Nanjing and at other sites of China.




Table 1 Statistical summary of the surface aerosol optical properties in Nanjing

| Factors | Max | Min | Mean±SD | Median |
|---|---|---|---|---|
| 550 nm AAC (Mm$^{-1}$) | 230.648 | 1.439 | 29.615±20.454 | 24.572 |
| 550 nm SC (Mm$^{-1}$) | 2493.092 | 20.673 | 338.275±228.078 | 284.379 |
| 550 nm Bsp (Mm$^{-1}$) | 300.101 | 1.401 | 44.257±27.396 | 38.206 |
| 550 nm EC (Mm$^{-1}$) | 2643.101 | 31.186 | 381.958±252.271 | 321.679 |
| 550 nm SSA | 0.988 | 0.404 | 0.901±0.049 | 0.908 |
| 550 nm ASP | 0.908 | 0.118 | 0.571±0.088 | 0.582 |
| 470/660 nm AAE | 3.256 | 0.145 | 1.583±0.228 | 1.592 |
| 450/635 nm SAE | 3.344 | 0.162 | 1.320±0.407 | 1.317 |

AAC: Aerosol absorption coefficient
SC: Aerosol scattering coefficient
Bsp: Aerosol back scattering coefficient
SSA: Aerosol single scattering albedo
ASP: Aerosol asymmetry parameter
AAE: Ångström exponent of absorbing aerosols
SAE: Ångström exponent of scattering aerosols

Table 2 Seasonal mean±SD of the surface aerosol optical properties in Nanjing

| Factors | MAM | JJA | SON | DJF |
|---|---|---|---|---|
| 550 nm AAC (Mm$^{-1}$) | 26.954±18.632 | 19.653±15.689 | 33.474±19.686 | 37.958±21.892 |
| 550 nm SC (Mm$^{-1}$) | 318.998±202.264 | 340.865±226.151 | 294.624±200.052 | 385.137±255.282 |
| 550 nm Bsp (Mm$^{-1}$) | 42.995±23.580 | 36.990±25.067 | 38.684±23.017 | 54.786±30.974 |
| 550 nm EC (Mm$^{-1}$) | 341.279±209.315 | 370.236±248.125 | 351.887±244.267 | 422.569±273.565 |
| 550 nm SSA | 0.915±0.043 | 0.933±0.049 | 0.874±0.053 | 0.890±0.040 |
| 550 nm ASP | 0.553±0.086 | 0.638±0.069 | 0.566±0.079 | 0.540±0.083 |



| | | | | |
|---|---|---|---|---|
| 470/660 nm AAE | 1.571±0.172 | 1.488±0.263 | 1.524±0.277 | 1.701±0.156 |
| 450/635 nm SAE | 1.097±0.320 | 1.337±0.428 | 1.544±0.352 | 1.235±0.383 |




Table 3 The aerosol optical properties both in Nanjing and at other sites of China

| Site | Period | AAC (Mm⁻¹) | SC (Mm⁻¹) | ASP | SSA | | Method | References |
|---|---|---|---|---|---|---|---|---|
| Nanjing (urban) | 2014.3-2016.2 | 29.6 (550 nm) | 338.3 (550 nm) | 0.57 (550 nm) | 0.9 | (550 nm) | [a] AE-31 [b] Aurora 3000 | This stuty |
| Beijing (urban) | 2005-2006 | 56 (532 nm) | 288 (525 nm) | / | 0.8 | (525 nm) | [c] AE-16 [d] M9003 | He et al. (2009) |
| Beijing (rural) | 2003-2005 | 17.5 (525 nm) | 174.6 (525 nm) | / | 0.88 | (525 nm) | [a] AE-31 [d] M9003 | Yan et al. (2008) |
| Xi'an (urban) | 2009 | / | 525 (520 nm) | / | / | | [e] Auroral 1000 | Cao et al. (2012) |
| Chengdu (urban) | 2011 | 96 (532 nm) | 456 (520 nm) | | 0.82 | | [a] AE-31 [f] Aurora 1000G | Tao et al. (2014) |
| Wuhan (urban) | 2009.12-2014.03 | 119 (520 nm) | 377 (550 nm) | / | 0.73 | (520 nm) | [a] AE-31 [g] Model 3563 | Gong et al. (2015) |
| Xinken (rural) | 2004.10-2011.05 | 70 (550 nm) | 333 (550 nm) | / | 0.83 | (550 nm) | [h] MAAP [g] Model 3563 | Cheng et al. (2008) |
| Tongyu (rural) | Spring, 2010 Spring, 2011 | 7.61 (520 nm) 7.01 (520 nm) | 89.2 (520 nm) 85.3 (520 nm) | / / | 0.9 | (520 nm) | [a] AE-31 [b] Aurora 3000 | Wu et al. (2012) |
| Nanjing (suburban) | 2011.03-04 | 28.1 (532 nm) | 329.3 (550 nm) | / | 0.89 | (532 nm) | [i] PASS [d] Model 3563 | Yu et al. (2016) |
| Shanghai (ruban) | 2010.12-2011.03 | 66 (532 nm) | 293 (532 nm) | / | 0.81 | (532 nm) | [a] AE-31 [g] Model 3563 | Xu et al. (2012) |
| Shouxian (rural) | 2008.5-12 | 29 (550 nm) | 401 (550 nm) | / | 0.92 | (550 nm) | [j] Model PSAP [g] Model 3563 | Fan et al. (2010) |
| Lanzhou (urban) | Winter 2001, 2002 | / | 226 (550 nm) | / | / | | [d] Model 3563 | Zhang et al. (2004) |
| Panyu (urban) | Spring and winter, 2008 | 84.03 and 188.8 (532 nm) | / | / | / | | [a] AE-31 | Wu et al. (2013) |
| Dongguan (suburban) | Spring and winter, 2008 | 47.1 and 95.53 (532 nm) | / | / | / | | [a] AE-31 | Wu et al. (2013) |





| | | | | | | | | |
|---|---|---|---|---|---|---|---|---|
| Maofengshan (Rural) | Spring and winter, 2008 | 26.45 and 28.77 (532 nm) | / | / | / | / | [a] AE-31 | Wu et al. (2013) |
| Yongxing Island | Spring and winter, 2008 | 7.21 and 8.37 (532 nm) | / | / | / | / | [a] AE-31 | Wu et al. (2013) |

[a] Seven channels Aethalomter (model AE-31, Magee Scientific, USA)
[b] Three wavelength integrating Nephelometer (Model Aurora 3000, Australia)
[c] Aethalometer AE16
[d] Nephelometer M9003
[e] Integrating Nephelometer (Model Aurora 1000)
[f] Integrating Nephelometer (Model Aurora 1000G)
[g] Integrating Nephelometer (Model 3563, TSI, USA)
[h] Multi-angle Absorption Photometer (MAAP, Thermo, Inc., Waltham, MA USA,
Model 5012)
[i] Photo acoustic Soot Spectrometer (PASS 1, DMT, USA)
[j] Particle/SootAbsorption Photometer





**780      Figures:**

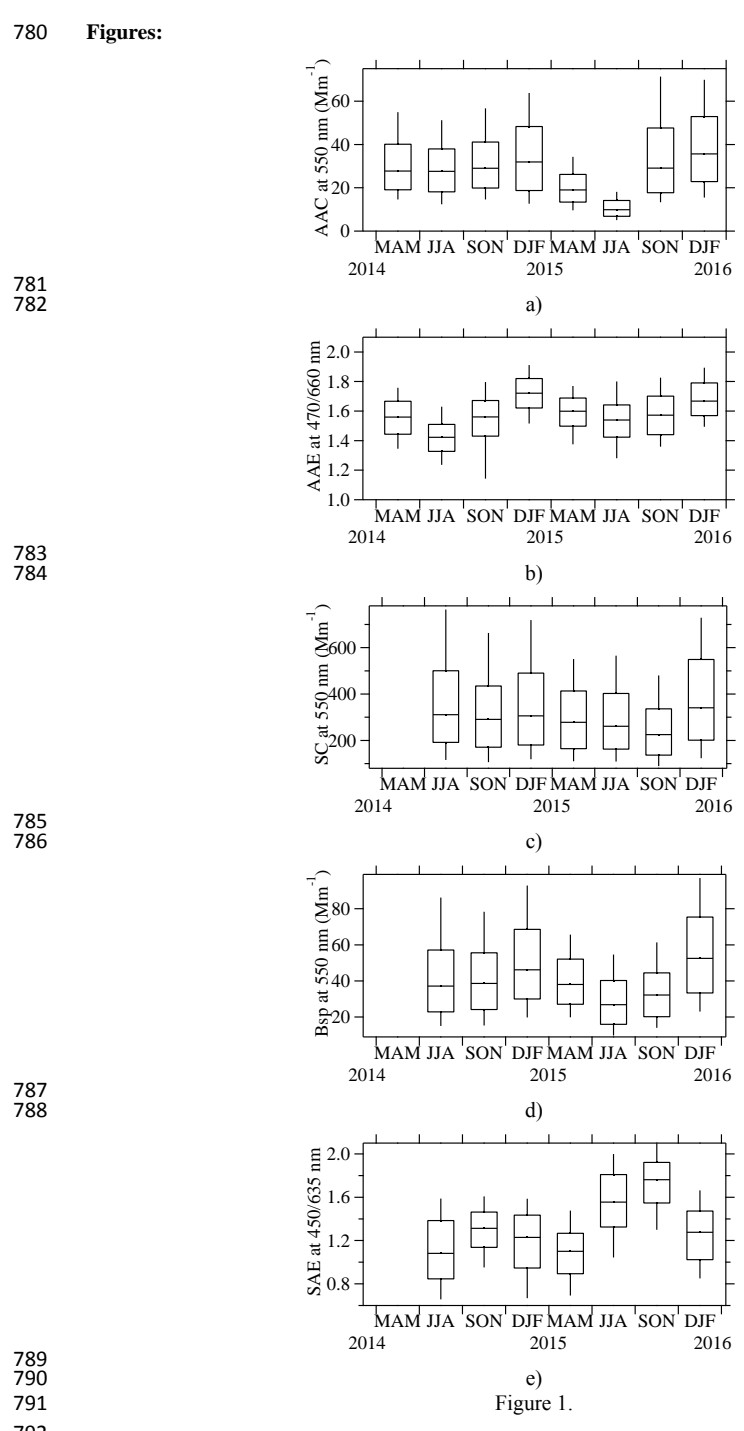

a)

b)

c)

788                                                              d)

e)
Figure 1.

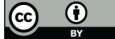



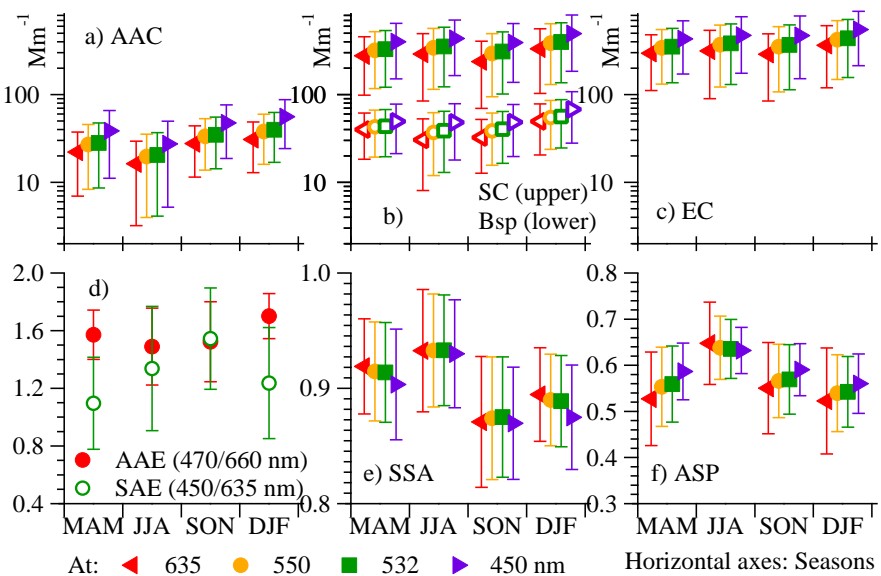


Figure 2.


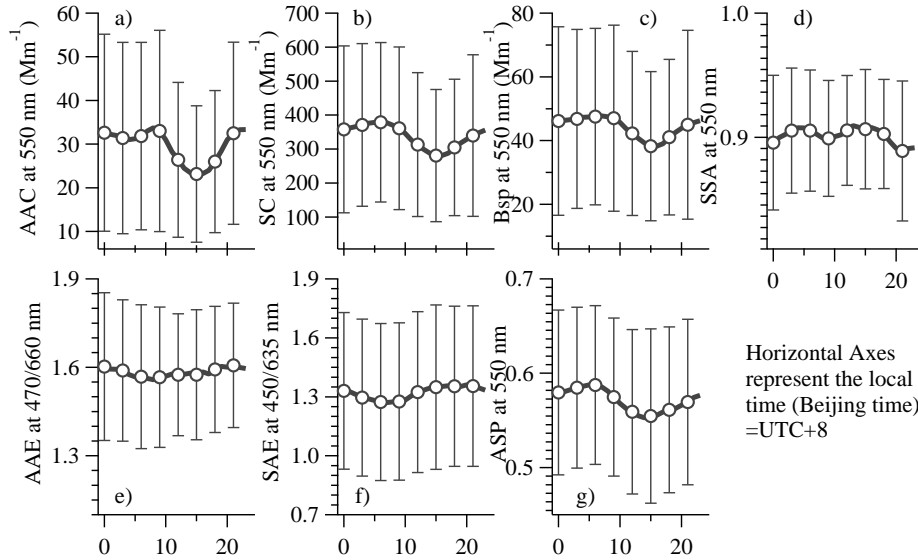


797                      Figure 3


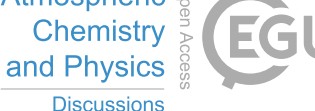




Figure 4

a) 804 805 b)
c) 808 809 d)
Figure 5



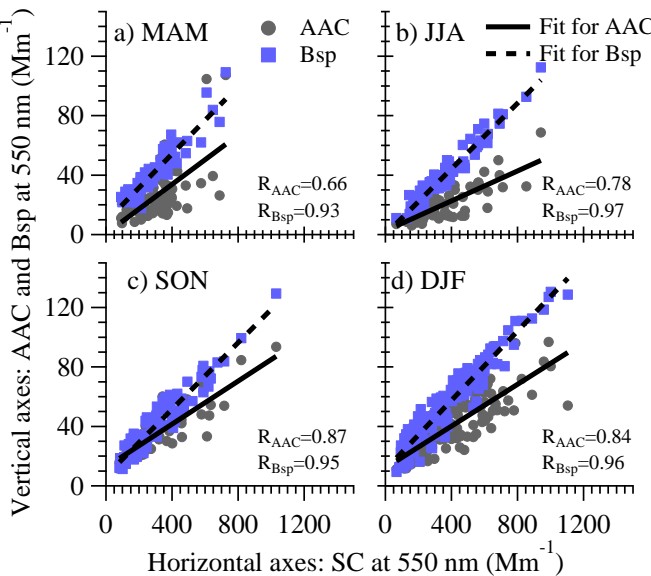


813                                        Figure 6


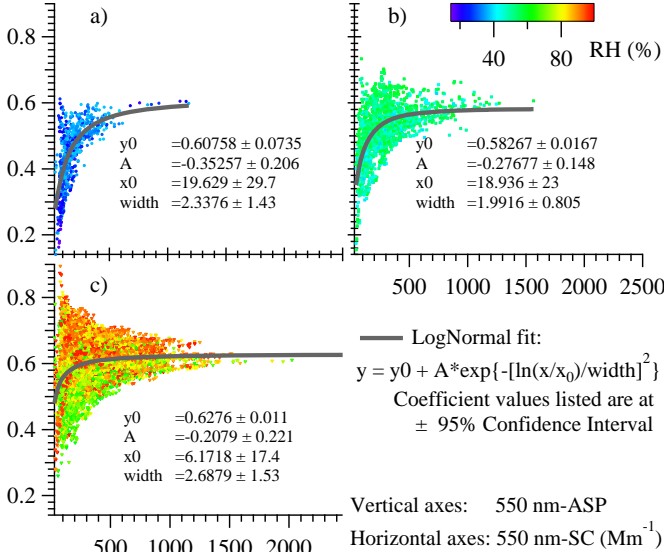


816                                        Figure 7






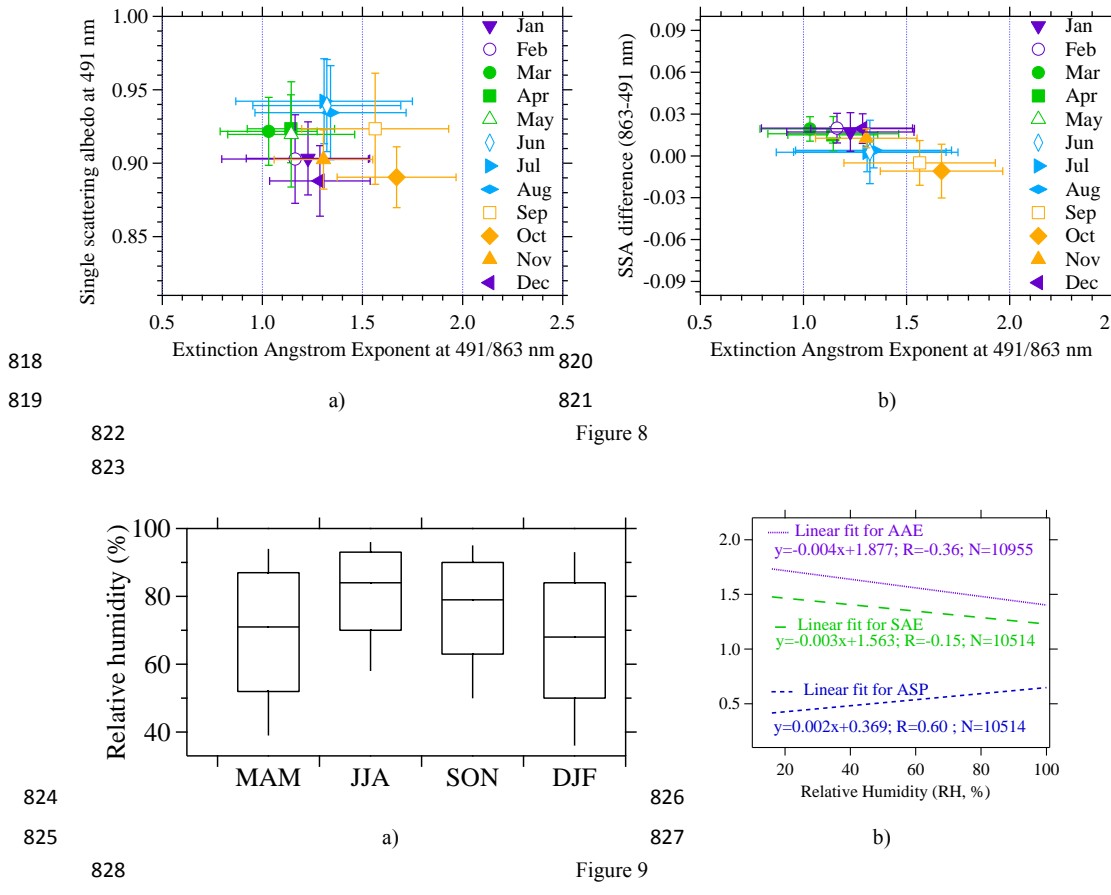

Figure 8

Figure 9





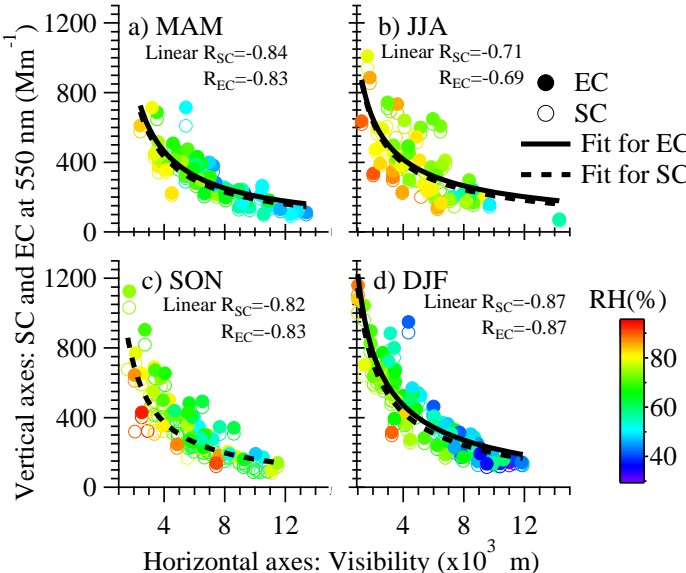


830                                   Figure 10


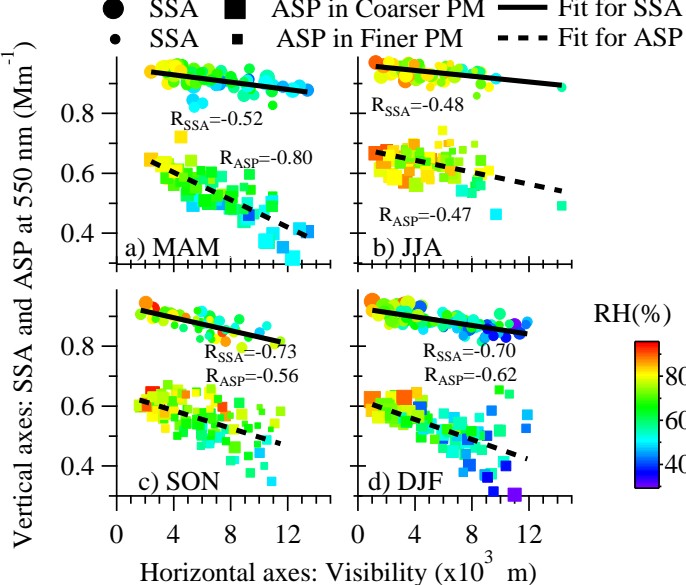


833                                   Figure 11






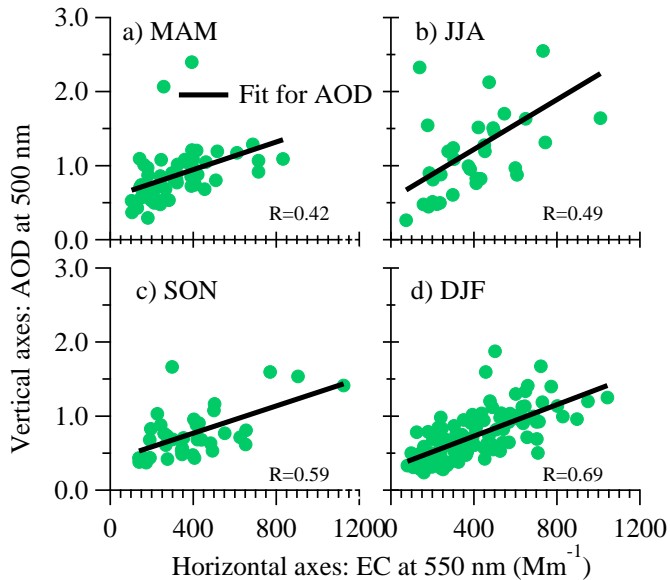


836                                          Figure 12