# Peer review of "The surface aerosol optical properties in urban areas of Nanjing,"

_Atmospheric Chemistry and Physics, 2016_

## Referee Comment (RC1) · Anonymous Referee #1 · 7 Sep 2016

This manuscript presents surface aerosol optical properties based on two years of measurements by an Aethalometer and a Nephelometer at an urban site in Nanjing. Authors analyzed their seasonal to diurnal variability and discussed their relationships with relative humidity, wind direction, and visibility. Overall this manuscript is clearly written. Its study provides an important observation-based characterization of aerosol properties over the study area. Meanwhile, manuscript could be further improved in a few aspects as I comment below.

Major comments:

1. The first impression from reading through the manuscript is that it presents so many numbers that readers could easily get lost. It is especially the case when it presents

the literature survey in the Line 70 – 91. I would suggest summarize those number in a way that readers can better grab readers's interest, for example, presenting them in a Table. This is also related to my major comment #2 as below.

2. It seems to me the content in the paragraph 345 – 367 closely relates to that in the Line 70 – 91. Why joint them together and put relevant numbers in Table 3?

3. Another major comments is about the RH effect on optical properties of hydrophilic particles. While water vapor is an important factor affecting the optical properties, this manuscript tends to overstate its role in the seasonal variability of aerosol optical properties. For example, higher SC, smaller AAE, and SAE in summer season are extensively attributed to the higher RH value. However, the seasonality of surface aerosol property is also influenced by the variability of PBL height, dry/wet deposition, and aerosol emissions. The roles of these factors are rarely discussed in the manuscript.

Specific comments:

52–56: The radiative forcing results should be updated to the latest IPCC report, i.e., the 5th AR.

200: Are the measurements from this single site able to represent the "urban area of Nanjing"? Please justify.

221: "moisture absorption growing" –> "water-uptake growth" or "hydroscopicity"

273–274: It is neither persuasive nor clear to the reviewer to say "SSA is also large in afternoon possibly because the dilution effect of well developed boundary layer on scattering aerosol is weaker than that on absorbing aerosols." Please justify or present more clearly.

Technical corrections:

17: than –> than aerosols

58: is mostly resulted from –> mostly results from

61: are corrected based on –> are based on

63: among –> among countries in

69–70: "Uncertainties in … the rest of the world." –> "Uncertainties of the aerosol radiative forcing and corresponding climate effects in these regions might be much larger than those of the rest of the world."

188: were directly –> , which were directly

201: the scattering aerosols' optical properties –> aerosol scattering properties

207: might be mostly resulted from –> might result from

375: 0.87 –> 0.78

---

## Referee Comment (RC2) · Anonymous Referee #3 · 3 Oct 2016

**General comments:**

This manuscript presents very profound analyses with regard to the aerosol properties in Nanjing, China, using two years of surface-level aerosol observations in combination of coincident meteorological measurements. In particular, the scattering and absorbing properties of aerosols are investigated by linking with RH and visibility, on both diurnal and seasonal scales. The results obtained here gain great insight into how aerosol varies with meteorology, and how scattering aerosol can be differentiated from absorbing aerosols in the highly polluted but populous region of YRD. In this revision, the authors have put more efforts and make thorough changes per the referee's comments, both grammatically and scientifically, which increases its readability. Therefore, I recommend this work be published in ACP after the following concerns have been adequately addressed.

**Major points for consideration:**
1. What is the difference between "Introduction: lines 63-102" and "description with regard to Table 3 (lines 345-367)", which seems a little redundant. Most of the cited literatures are the same, i recommend the authors delete or shorten the paragraph or rephrase it in the Introduction.

2. Line 475-476:  "… three-wavelength integrating Nephelometer (Aurora 3000, Australia) in urban area of Nanjing from Mar 2014 to Feb 2016" contracts with previous statements "because the measurements of Aurora 3000 started from June 2014.", the authors can choose to modify the text to make them consistent with each other.

**Minor points for consideration:**
Abstract: "… the regions around." -> "the surrounding regions."

Abstract: Line 35: "It" refers to ? clarify it.

Line 42: More work can be cited here: "...or global climate changes (e.g., Forster et al., 2007, Rosenfeld et al., 2008; Li et al., 2011; Wang et al., 2014; Guo et al., 2016)."

References:

Rosenfeld, D., Lohmann, U., Raga, G.B., O'Dowd, C.D., Kulmala, M., Fuzzi, S., Reissell, A., Andreae, M.O.: Flood or drought: how do aerosols affect precipitation? Science 321, 5894, 1309-1313. 2008.

Qian, Y., Gong, D.Y., Fan, J.W., Leung, L.R., Bennartz, R., Chen, D.L., Wang, W.G.: Heavy pollution suppresses light rain in China: Observations and modeling. J. Geophys. Res. 114, D00K02, doi:10.1029/2008jd011575. 2009.

Li, Z., Li, C., Chen, H., et al.: East Asian Studies of Tropospheric Aerosols and their Impact on Regional Climate (EAST-AIRC): An overview. J. Geophys. Res. 116, D7, doi:10.1029/2010jd015257.2011.

Wang, Y., Wang, M., Zhang, R., et al., 2014. Assessing the effects of anthropogenic aerosols on Pacific storm track using a multiscale global climate model. Proceedings of the National Academy of Sciences 111, 19, 6894-6899.

Guo, J., M. Deng, S. S. Lee, F. Wang, Z. Li, P. Zhai, H. Liu, W. Lv, W. Yao, and X. Li: Delaying precipitation and lightning by air pollution over the Pearl River Delta. Part I: Observational analyses, J. Geophys. Res. Atmos., 121, 6472–6488, doi:10.1002/2015JD023257.2016.

Line 58: Grammar error: "The bias is mostly resulted from" -> "The bias mostly results from"

Line 60: before "The uncertainty could be substantially", the authors may add one more sentence here as follows: "In addition, the diurnal variability of aerosol properties has been suggested to another major factors leading to such large biases (e.g., Xu et al., AE 2016).

Reference:

Xu H., J.P. Guo, X. Ceamanos, Roujean J.L., M. Min, D. Carrer: On the influence of the diurnal variations of aerosol content to estimate direct aerosol radiative forcing using MODIS data, Atmospheric Environment, 141, 186–196. doi: 10.1016/j.atmosenv.2016.06.067. 2016.

Line 64: "trace gases (Zhang et al., 2009)." -> "trace gases (e.g., Guo et al., 2009; Zhang et al., 2009; Che et al., 2015)."

Rerences:

Guo, J.P., X. Zhang, H. Che, S. Gong, X. An, C.X. Cao, J. Guang, H. Zhang, Y.Q. Wang, X.C. Zhang, P. Zhao, X.W. Li: Correlation between PM Concentrations and Aerosol Optical Depth in Eastern China, Atmospheric Environment, 43(37): 5876-5886. 2009.

Xin, J., Wang, Y., Pan, Y., et al.: The Campaign on Atmospheric Aerosol Research Network of China: CARE-China. Bulletin of the American Meteorological Society 96, 7, 1137-1155, doi:10.1175/BAMS-D-14-00039.1. 2014.

Che, H. Z., Zhang, X. Y., Xia, X., et al.: Ground-based aerosol climatology of China: aerosol optical depths from the China Aerosol Remote Sensing Network (CARSNET) 2002–2013, Atmos. Chem. Phys., 15, 7619–7652, 2015.

Line 81: center -> central

Line 100-102: "Our ultimate goals are to reduce uncertainties in estimating aerosol radiative forcing and climate effect and to improve forecast accuracy of visibility" SHOULD BE CHANGED because this paper does not contain anything on radiative forcing or climate effect.

Line 106: "Methodologies" -> "Data and methodologies"

Line 117: "To make a brief comparison.."    the authors can add more words to clarify "..comparison with what?? "

Line 126: what kind of "Meteorological data", please give a detailed information here.

Line 192: "Eq. 8~10" -> "Eqs. 8~10"

Line 207: "Thus, ..., lower boundary height and less rainfall" -> "Thus,.... lower boundary height (Guo et al., 2016) and less rainfall.."

Reference:

Guo, J., Miao, Y., Zhang, Y., Liu, H., Li, Z., Zhang, W., He, J., Lou, M., Yan, Y., Bian, L., and Zhai, P.: The climatology of planetary boundary layer height in

China derived from radiosonde and reanalysis data, Atmos. Chem. Phys. Discuss., doi:10.5194/acp-2016-564, 2016.

Line 209: "RH." -> "hygroscopic growth of aerosol caused by higher RH.."

Line 211: "RH" -> "relatively higher RH"

Line 221: "if the moisture absorption growing" you mean "hygroscopic growth"?

Line 224-227: Deleted "which might somewhat relate to a difference in RH in these two years", and add "The observed RH difference in these two years at least partly accounts for the variation of aerosol absorption coefficient and scattering coefficient as well as their sizes (or sth describing the aerosol properties in previous text)" just following "…69.03 in SON)."

Line 230: "..at four wavelengths," is not consistent with text in the first paragraph of Section 3 which states that only one wavelength (i.e., 550nm) will be discussed. So the authors should modify either the statements here or the former statements in section 3.

Line 260: Delete "are"

Line: 290: "of" is omitted before the statement " the total samplings..."

Line 372: " obviously" -> " obvious"

Lines 372-374: " It is obvious.." should be placed behind " The linear correlation coefficient varies from 0.93 to 0.97 for SC ... AAC in urban Nanjing

."Lines 377-378: Improper citation: Behind " in the same season in 2011..", add "which agrees well with Yu et al. (2016)"

Figure 8: Figure caption should better be revised to reflect the monthly variation for the observed aerosol properties.

Figure 9b: the legend for the linear fit for SEA should be dashed line

---

## Referee Comment (RC3) · Anonymous Referee #4 · 7 Dec 2016

Review of "The surface aerosol optical properties in urban areas of Nanjing, west Yangtze River Delta of China" by Zhuang et al., 2016.

This manuscript provides an in depth analysis of aerosol optical properties spanning several years using surface observations of aerosol optical properties in Nanjing, China. While the results presented in the manuscript are valuable to the community, the organization and presentation of the manuscript is overwhelming. Therefore, I recommend publication with major revision.

Major Comments: 1. As previously mentioned, presentation of results in the manuscript is overwhelming. Specifically, listing Bsp, AAC, EC, etc., values over and over again is very confusing. These numbers are listed in tables, and I suggest that the authors refer

to them that way, rather than listing in the manuscript. 2. Observations like this are of great benefit to the community, but I feel the seasonal and diurnal observations are of great importance to the modeling community. I think these findings could be highlighted more as a useful resource to aerosol modelers and discussed in the conclusion. It would be particularly valuable if the authors could parse out the optical properties at a monthly scale, which could be directly read by a model, for example.

Minor Comments: 1. Lines 81 – 102. It would be easier to interpret and compare values presented in this paragraph as a table. 2. Line 29. What is Bsp? 3. Line 30. What is AE? 4. Line 35. What "could be further deteriorated"? 5. Lines 38 – 39. Please provide a reference. 6. Line 42. This sentence requires several more references. 7. Lines 59 – 60. How can the bias in Zhuang et al., 2013a be explained by Holler at el al., 2003? One study was 10 years prior to the other. 8. Lines 69 – 70. Please provide a reference. 9. Lines 133. What does ATN stand for?

Technical Comments: 1. Line 56. Radiative.

---

## Author Comment (AC1) · 15 Dec 2016

Reviewer 1:

Anonymous Referee 1

This manuscript presents surface aerosol optical properties based on two years of measurements by an Aethalometer and a Nephelometer at an urban site in Nanjing. Authors analyzed their seasonal to diurnal variability and discussed their relationships with relative humidity, wind direction, and visibility. Overall this manuscript is clearly written. Its study provides an important observation-based characterization of aerosol

properties over the study area. Meanwhile, manuscript could be further improved in a few aspects as I comment below.

To Referee 1:

Dear reviewer, thank you very much for reviewing the manuscript and providing us the constructive comments and suggestions on our study. We have learned a lot from your advices. With respect to your comments, necessary revisions of the paper have been made. We will response to your comments carefully point by point; details of the revisions can be referred to the revised version of the manuscript. Relevant changes of the revised manuscript are listed in the last page.

Major comments: 1. The first impression from reading through the manuscript is that it presents so many numbers that readers could easily get lost. It is especially the case when it presents the literature survey in the Line 70 – 91. I would suggest summarize those number in a way that readers can better grab readers's interest, for example, presenting them in a Table. This is also related to my major comment 2 as below.

R: Thank you very much for your suggestion. We agree with you that so many aerosol optical properties here would make readers confuse and it would be more readable by presenting them in a Table. The numbers listed in these lines are mainly aimed at introducing current states and progresses in researching observed based aerosol optical properties over China. In view of most of these numbers are deeply discussed in the text and listed in the table, Line 70-92 have been rephrased and shortened in revised manuscript to avoid duplication.

2. It seems to me the content in the paragraph 345 – 367 closely relates to that in the Line 70 – 91. Why joint them together and put relevant numbers in Table 3?

R: Thank you for your question. As mentioned in comment 1 above, the numbers listed in Line 70-91 are aimed at introducing research progresses of the aerosol optical properties. To make comparisons of the surface aerosol optical properties between

urban area of Nanjing and the other sites over China, these numbers as well as the results in this study are listed together in a table to make readers easier to read. And Line 345-367 are the corresponding statements of the table. Meanwhile, Line 70-91 in Introduction have been shorten in the revised version of the manuscript to avoid duplication.

3. Another major comments is about the RH effect on optical properties of hydrophilic particles. While water vapor is an important factor affecting the optical properties, this manuscript tends to overstate its role in the seasonal variability of aerosol optical properties. For example, higher SC, smaller AAE, and SAE in summer season are extensively attributed to the higher RH value. However, the seasonality of surface aerosol property is also influenced by the variability of PBL height, dry/wet deposition, and aerosol emissions. The roles of these factors are rarely discussed in the manuscript.

R: In summer, both trace gases and particulate matters have lower emission rates as suggested by Zhang et al. (2009). Furthermore, PBL height and precipitation mostly have larger values in this season than those in other seasons. Thus, these three factors would likely result in smaller aerosol loadings in summer. However, SC in summer is larger than that in spring and fall and it was thought to possibly due to the effects of RH. Because: Zhang et al. (2015) indicated that SC and Bsp in YRD would increase by 50

References:

Zhang, Q., Streets, D. G., Carmichael, G. R., He, K. B., Huo, H., Kannari, A., Klimont, Z., Park, I. S., Reddy, S., Fu, J. S., Chen, D., Duan, L., Lei, Y., Wang, L. T., and Yao, Z. L.: Asian emissions in 2006 for the NASA INTEX-B mission, Atmos. Chem. Phys., 9, 5131–5153, doi:10.5194/acp-9-5131-2009, 2009.

Zhang, L., Sun, J. Y., Shen, X. J., Zhang, Y. M., Che, H., Ma, L. Q., Zhang, Y. W., Zhang, X. Y., and Ogren, J. A.: Observations of relative humidity effects on aerosol light scattering in the Yangtze River Delta of China, Atmos. Chem. Phys., 15, 8439–

8454, 2015.

Zhuang, B. L., Wang, T. J., Li, S., Liu, J., Talbot, R., Mao, H. T., Yang, X. Q., Fu, C. B., Yin, C. Q., Zhu, J. L., Che, H. Z., and Zhang, X. Y.: Optical properties and radiative forcing of urban aerosols in Nanjing, China, Atmos. Environ., 83, 43–52, 2014a.

Specific comments: 52–56: The radiative forcing results should be updated to the latest IPCC report, i.e., the 5th AR.

R: Thank you for your suggestion. The radiative forcing of all aerosols and black carbon aerosol from 5th IPCC have been used in the revised manuscript.

200: Are the measurements from this single site able to represent the "urban area of Nanjing"? Please justify.

R: Thank you for your question. We believe that the site can represent the urban area of Nanjing. Firstly, it's located in the down town area of Nanjing. And more important, it is built on the roof of a 79.3 m-tall building instead of the surface to avoid the local influences (such as: the block) below the urban canopy as far as possible. Second, there almost have no higher buildings around and there are no industrial pollution sources within a 30 km radius around the site. Third, due to its good representation, trace gases (such as: Hgg, $CO_2$, CO, $O_3$, $NO_x$, $NO_y$) and aerosols (such as: Hgp, BC, PM) in urban areas of Nanjing were well observed regularly and in seriously polluted episodes. Some of them have been published as followed:

Han, Y., Y. H. Wu, T. J. Wang, B. L. Zhuang, S. Li, K. Zhao (2015), Impacts of elevated-aerosol-layer and aerosol type on the correlation of AOD and particulate matter with ground-based and satellite measurements in Nanjing, southeast China, Science of the Total Environment, 532, 195-207, doi: 10.1016/j.scitotenv.2015.05.136.

Han, Y., Y. H. Wu, T. J. Wang, C. B. Xie, K. Zhao, B. L. Zhuang, S. Li (2015), Characterizing a persistent Asian dust transport event: Optical properties and impact on air quality through the ground-based and satellite measurements over Nanjing, China,

Atmospheric Environment, 115, 304-316, doi: 10.1016/j.atmosenv.2015.05.048.

Huang, X. X., T. J. Wang, R. Talbot, M. Xie, H. T. Mao, S. Li, B. L. Zhuang, X. Q. Yang, C. B. Fu, J. L. Zhu, X. Huang, R. Y. Xu (2015), Temporal characteristics of atmospheric $CO_2$ in urban Nanjing, China, Atmospheric Research, 153, 437-450, doi:10.1016/j.atmosres.2014.09.007.

Li, S., T. J. Wang, M. Xie, Y. Han, B. L. Zhuang (2015), Observed aerosol optical depth and angstrom exponent in urban area of Nanjing, China, Atmospheric Environment, 123, 350-356, doi: 10.1016/j.atmosenv.2015.02.048.

Zhu, J., T. J. Wang, R. Talbot, H. Mao, X. Yang, C. Fu, J. Sun, B. L. Zhuang, S. Li, Y. Han, M. Xie (2014), Characteristics of atmospheric mercury deposition and size-fractionted particulate mercury in urban Nanjing, China, Atmos. Chem. Phy., 14, 2233-2244.

Zhu, J., T. Wang, R. Talbot, H. Mao, C. Hall, X. Yang, C. Fu, B. L. Zhuang, S. Li, Y. Han, X. Huang (2012), Characteristics of atmospheric Total Gaseous Mercury (TGM) observed in urban Nanjing, China, Atmos. Chem. Phys., 12, 12103-12118.

Zhuang, B. L., T. J. Wang, J. Liu, Y. Ma, C. Q. Yin, S. Li, M. Xie, Y. Han, J. L. Zhu, X. Q. Yang, C. B. Fu (2015), Absorption coefficient of urban aerosol in Nanjing, west Yangtze River Delta, China, Atmos. Chem. Phy., 15, 13633-13646.

Zhuang, B. L., T. J. Wang, J. Liu, S. Li, M. Xie, X. Q. Yang, C. B. Fu, J. N. Sun, C. Q. Yin, J. B. Liao, J. L. Zhu, Y. Zhang (2014), Continuous measurement of black carbon aerosol in urban Nanjing of Yangtze River Delta, China, Atmos. Environ., 89, 415-424.

Zhuang, B. L., T. J. Wang, S. Li, J. Liu, R. Talbot, H. T. Mao, X. Q. Yang, C. B. Fu, C. Q. Yin, J. L. Zhu, H. Z. Che, X. Y. Zhang (2014), Optical properties and radiative forcing of urban aerosols in Nanjing, China, Atmos. Environ., 83, 43-52.

221: "moisture absorption growing" –> "water-uptake growth" or "hydroscopicity"

[Figure]

R: It has been changed to "hygroscopic growth" in revised manuscript.

273–274: It is neither persuasive nor clear to the reviewer to say "SSA is also large in afternoon possibly because the dilution effect of well developed boundary layer on scattering aerosol is weaker than that on absorbing aerosols." Please justify or present more clearly.

R: Thank you for your comments. The sentences have been rephrased in the revised manuscript to make them more clearly.

Technical corrections:

17: than –> than aerosols

R: Corrected.

58: is mostly resulted from –> mostly results from

R: Corrected.

61: are corrected based on –> are based on

R: Corrected.

63: among –> among countries in

R: Corrected.

69–70: "Uncertainties in . . . the rest of the world." –> "Uncertainties of the aerosol radiative forcing and corresponding climate effects in these regions might be much larger than those of the rest of the world."

R: Corrected.

188: were directly –> , which were directly

R: Corrected.
201: the scattering aerosols' optical properties –> aerosol scattering properties

R: Corrected.

207: might be mostly resulted from –> might result from

R: Corrected.

375: 0.87 –> 0.78

R: Corrected.

Relevant changes in revised manuscript:

Author affiliations: 1 and 3 are merged together. The last one is changed to "Department of Climate and Space Sciences and Engineering, University of Michigan, Ann Arbor, Michigan, USA"

In third paragraph of Introduction: Introductions on the aerosol optical properties in lines 70-91 in original manuscript are rephrased and shortened to avoid duplication and confusing, based on the reviewers' comments and suggestions.

In first paragraph of Section 3.2: To make the text more readable, frequency values are listed in a new table (Table 3 in revised manuscript) and corresponding statement has been rephrased according to reviewers' comments.

In fourth paragraph of Section 3.4: Add more discussion on the effect of RH on the aerosol optical properties.

In third and fourth paragraph in Conclusion: According to reviewers' comments, discussions on the importance of the aerosol optical properties seasonal and diurnal variations have been added. Additionally, frequency analysis was shortened.

In Acknowledgements: The foundation number was changed in revised manuscript. References: References listed and recommended in reviewers' comments were cited and listed in References section in revised manuscript.

In Figure captions: Fig. 8's caption was changed according to reviewers' comments.

Tables: Add a new table (Table 3) to summary the frequencies of the aerosol optical properties. Add a table caption in Table captions section. Table 3 in original manuscript is changed to Table 4.

Figures: Redraw Fig. 9b based on reviewers' comments.

Others: Correct the grammar, cite more references and re-organize some sentences throughout the manuscript according to reviewers' comments.

Please also note the supplement to this comment:
http://www.atmos-chem-phys-discuss.net/acp-2016-420/acp-2016-420-AC1-supplement.pdf

---

## Author Comment (AC2) · 15 Dec 2016

The comment was uploaded in the form of a supplement as follow:

General comments:

This manuscript presents very profound analyses with regard to the aerosol properties in Nanjing, China, using two years of surface-level aerosol observations in combination of coincident meteorological measurements. In particular, the scattering and absorbing properties of aerosols are investigated by linking with RH and visibility, on both diurnal and seasonal scales. The results obtained here gain great insight into how aerosol varies with meteorology, and how scattering aerosol can be differentiated from absorbing aerosols in the highly polluted but populous region of YRD. In this revision, the authors have put more efforts and make thorough changes per the referee's comments, both grammatically and scientifically, which increases its readability. Therefore, I recommend this work be published in ACP after the following concerns have been adequately addressed.

To Referee 3:

Dear reviewer, thank you very much for reviewing the manuscript and providing us the constructive comments and suggestions on our study. We have learned a lot from your advices. With respect to your comments, necessary revisions of the paper have been made. We will response to your comments carefully point by point; details of the revisions can be referred to the revised version of the manuscript.

Relevant changes of the revised manuscript are listed in the last page.

Major points for consideration:

1. What is the difference between "Introduction: lines 63-102" and "description with regard to Table 3 (lines 345-367)", which seems a little redundant. Most of the cited literatures are the same, I recommend the authors delete or shorten the paragraph or rephrase it in the Introduction.

R: Thank you very much for your suggestion. We agree with you that it indeed has a little redundant. The numbers listed in lines 63-102 are mainly aimed at introducing current states and progresses in researching observed based aerosol optical properties over China. In view of these numbers mostly being discussed in the text, lines 63-102 have been shortened in revised manuscript to avoid duplication.

2. Line 475-476: ". . . three-wavelength integrating Nephelometer (Aurora 3000, Australia) in urban area of Nanjing from Mar 2014 to Feb 2016" contracts with previous statements "because the measurements of Aurora 3000 started from June 2014.", the authors can choose to modify the text to make them consistent with each other.

R: Thank you for your suggestion. According to you advice, Words "from Mar 2014 to Feb 2016" in the sentence have been deleted to make the context consistency in revised version of the manuscript.

Minor points for consideration:

Abstract: ". . . the regions around." -> "the surrounding regions."

R: It has been corrected in revised manuscript.

Abstract: Line 35: "It" refers to ? clarify it.

R: It means "Atmospheric visibility". Necessary change has been made in revised manuscript.

Line 42: More work can be cited here: "...or global climate changes (e.g., Forster et al., 2007, Rosenfeld et al., 2008; Li et al., 2011; Wang et al., 2014; Guo et al., 2016)."

References:

Rosenfeld, D., Lohmann, U., Raga, G.B., O'Dowd, C.D., Kulmala, M., Fuzzi, S., Reissell, A., Andreae, M.O.: Flood or drought: how do aerosols affect precipitation? Science 321, 5894, 1309-1313. 2008.

Qian, Y., Gong, D.Y., Fan, J.W., Leung, L.R., Bennartz, R., Chen, D.L., Wang, W.G.: Heavy pollution suppresses light rain in China: Observations and modeling. J. Geophys. Res. 114, D00K02, doi:10.1029/2008jd011575. 2009.

Li, Z., Li, C., Chen, H., et al.: East Asian Studies of Tropospheric Aerosols and their Impact on Regional Climate (EAST-AIRC): An overview. J. Geophys. Res. 116, D7, doi:10.1029/2010jd015257.2011.

[Figure]

Wang, Y., Wang, M., Zhang, R., et al., 2014. Assessing the effects of anthropogenic aerosols on Pacific storm track using a multiscale global climate model. Proceedings of the National Academy of Sciences 111, 19, 6894-6899.

Guo, J., M. Deng, S. S. Lee, F. Wang, Z. Li, P. Zhai, H. Liu, W. Lv, W. Yao, and X. Li: Delaying precipitation and lightning by air pollution over the Pearl River Delta. Part I: Observational analyses, J. Geophys. Res. Atmos., 121, 6472–6488, doi:10.1002/2015JD023257.2016.

R: References listed above have been cited in revised manuscript.

Line 58: Grammar error:"The bias is mostly resulted from"-> "The bias mostly results from"

R: Corrected.

Line 60: before "The uncertainty could be substantially", the authors may add one more sentence here as follows: "In addition, the diurnal variability of aerosol properties has been suggested to another major factors leading to such large biases (e.g., Xu et al., AE 2016).

Reference:

Xu H., J.P. Guo, X. Ceamanos, Roujean J.L., M. Min, D. Carrer: On the influence of the diurnal variations of aerosol content to estimate direct aerosol radiative forcing using MODIS data, Atmospheric Environment, 141, 186–196. doi:10.1016/j.atmosenv.2016.06.067. 2016.

R: References listed above have been cited and corresponding statements have been included in revised manuscript.

Line 64: "trace gases (Zhang et al., 2009)." -> "trace gases (e.g., Guo et al., 2009; Zhang et al., 2009; Che et al., 2015)."

References:

[Figure]

Guo, J.P., X. Zhang, H. Che, S. Gong, X. An, C.X. Cao, J. Guang, H. Zhang, Y.Q. Wang, X.C. Zhang, P. Zhao, X.W. Li: Correlation between PM Concentrations and Aerosol Optical Depth in Eastern China, Atmospheric Environment, 43(37): 5876-5886. 2009.

Xin, J., Wang, Y., Pan, Y., et al.: The Campaign on Atmospheric Aerosol Research Network of China: CARE-China. Bulletin of the American Meteorological Society 96, 7, 1137-1155, doi:10.1175/BAMS-D-14-00039.1. 2014.

Che, H. Z., Zhang, X. Y., Xia, X., et al.: Ground-based aerosol climatology of China: aerosol optical depths from the China Aerosol Remote Sensing Network (CARSNET) 2002–2013, Atmos. Chem. Phys., 15, 7619–7652, 2015.

R: References listed above have been cited in revised manuscript.

Line 81: center -> central

R: Corrected.

Line 100-102: "Our ultimate goals are to reduce uncertainties in estimating aerosol radiative forcing and climate effect and to improve forecast accuracy of visibility" SHOULD BE CHANGED because this paper does not contain anything on radiative forcing or climate effect.

R: With respect to you suggestion, the sentence has been rephrased in revised manuscript.

Line 106: "Methodologies" -> "Data and methodologies"

R: Corrected.

Line 117: "To make a brief comparison.." the authors can add more words to clarify "..comparison with what?? "

R: It has been changed to "To make a brief comparison between surface and column aerosols"

Line 126: what kind of "Meteorological data", please give a detailed information here.

R: Thank you for your suggestion. Detailed information has been included.

Line 192: "Eq. 8 10" -> "Eqs. 8 10"

R: Corrected.

Line 207: "Thus, ..., lower boundary height and less rainfall" -> "Thus,.... lower boundary height (Guo et al., 2016) and less rainfall.."

Reference:

Guo, J., Miao, Y., Zhang, Y., Liu, H., Li, Z., Zhang, W., He, J., Lou, M., Yan, Y., Bian, L., and Zhai, P.: The climatology of planetary boundary layer height in China derived from radiosonde and reanalysis data, Atmos. Chem. Phys. Discuss., doi:10.5194/acp-2016-564, 2016.

R: Reference listed above has been cited in revised manuscript.

Line 209: "RH." -> "hygroscopic growth of aerosol caused by higher RH.."

R: Corrected.

Line 211: "RH" -> "relatively higher RH"

R: Corrected.

Line 221: "if the moisture absorption growing" you mean "hygroscopic growth"?

R: Yes. And the "moisture absorption growing" has been changed to "hygroscopic growth" in revised manuscript.

Line 224-227: Deleted "which might somewhat relate to a difference in RH in these two years", and add "The observed RH difference in these two years at least partly accounts for the variation of aerosol absorption coefficient and scattering coefficient as well as their sizes (or sth describing the aerosol properties in previous text)" just

following ". . .69.03 in SON)."

R: Thank you. The sentences have been rephrased based your suggestion.

Line 230: "..at four wavelengths," is not consistent with text in the first paragraph of Section 3 which states that only one wavelength (i.e., 550nm) will be discussed. So the authors should modify either the statements here or the former statements in section 3.

R: Based on your suggestion. Words "at four wavelengths " have been deleted in revised manuscript.

Line 260: Delete "are"

R: Delete.

Line: 290: "of" is omitted before the statement " the total samplings..."

R: Thanks. "of" has been added to the sentence.

Line 372: " obviously" -> " obvious"

R: Corrected.

Lines 372-374: " It is obvious.." should be placed behind " The linear correlation coefficient varies from 0.93 to 0.97 for SC ... AAC in urban Nanjing."

R: According to your suggestion. This sentence has been placed behind "The linear correlation coefficient varies from 0.93 to 0.97 for SC ... AAC in urban Nanjing."

Lines 377-378: Improper citation: Behind " in the same season in 2011..", add "which agrees well with Yu et al. (2016)"

R: The linear correlation coefficients between SC and AAC and between SC and Bsp in MAM in suburban Nanjing were carried out by Yu et al. (2016). Thus, the reference is cited behind: "...behind suburban Nanjing" So it's appropriate.

Figure 8: Figure caption should better be revised to reflect the monthly variation for the observed aerosol properties.

R: According to your comment, " the monthly variation " has been included in the Figure caption of Fig. 8.

Figure 9b: the legend for the linear fit for SEA should be dashed line

R: Fig. 9b has been redrawn based on your comment.

Relevant changes in revised manuscript:

Author affiliations: 1 and 3 are merged together. The last one is changed to "Department of Climate and Space Sciences and Engineering, University of Michigan, Ann Arbor, Michigan, USA"

In third paragraph of Introduction: Introductions on the aerosol optical properties in lines 70-91 in original manuscript are rephrased and shortened to avoid duplication and confusing, based on the reviewers' comments and suggestions.

In first paragraph of Section 3.2: To make the text more readable, frequency values are listed in a new table (Table 3 in revised manuscript) and corresponding statement has been rephrased according to reviewers' comments.

In fourth paragraph of Section 3.4: Add more discussion on the effect of RH on the aerosol optical properties.

In third and fourth paragraph in Conclusion: According to reviewers' comments, discussions on the importance of the aerosol optical properties seasonal and diurnal variations have been added. Additionally, frequency analysis was shortened.

In Acknowledgements: The foundation number was changed in revised manuscript.

References: References listed and recommended in reviewers' comments were cited and listed in References section in revised manuscript.

In Figure captions: Fig. 8's caption was changed according to reviewers' comments.

Tables: Add a new table (Table 3) to summary the frequencies of the aerosol optical properties. Add a table caption in Table captions section. Table 3 in original manuscript is changed to Table 4.

Figures: Redraw Fig. 9b based on reviewers' comments.

Others: Correct the grammar, cite more references and re-organize some sentences throughout the manuscript according to reviewers' comments.

Please also note the supplement to this comment:
http://www.atmos-chem-phys-discuss.net/acp-2016-420/acp-2016-420-AC2-supplement.pdf

---

## Author Comment (AC3) · 15 Dec 2016

Review of "The surface aerosol optical properties in urban areas of Nanjing, west Yangtze River Delta of China" by Zhuang et al., 2016.

This manuscript provides an in depth analysis of aerosol optical properties spanning several years using surface observations of aerosol optical properties in Nanjing, China. While the results presented in the manuscript are valuable to the community, the organization and presentation of the manuscript is overwhelming. Therefore, I rec-

ommend publication with major revision.

To Referee 4:

Dear reviewer, thank you very much for reviewing the manuscript and providing us the constructive comments and suggestions on our study. We have learned a lot from your advices. With respect to your comments, necessary revisions of the paper have been made. We will response to your comments carefully point by point; details of the revisions can be referred to the revised version of the manuscript.

Relevant changes of the revised manuscript are listed in the last page.

Major Comments:

1. As previously mentioned, presentation of results in the manuscript is overwhelming. Specifically, listing Bsp, AAC, EC, etc., values over and over again is very confusing. These numbers are listed in tables, and I suggest that the authors refer to them that way, rather than listing in the manuscript.

R: Thank you very much for your comment and suggestion. We agree with you that so many aerosol optical properties over and over again in the manuscript would make readers confuse and it would be more readable by presenting them in a Table. Lines 70-92 in Introduction are almost the same as lines 345-367 in Results and Discussions section. The numbers listed in former lines are mainly aimed at introducing the research progresses of observed aerosol optical properties over China while in latter lines: they are listed in the table and used to make comparisons with our observations. With respect to your suggestion, lines 70-92 have been rephrased and shortened in revised manuscript to avoid duplication and confusing. Additionally, a new table (Table 3 in revised manuscript) has been created to list the frequencies of the aerosol optical properties in (original) lines 288-293 to make them more readable. Further, lines 499-502 (original lines in old version of the manuscript) in Conclusions section have also been rephrased.

2. Observations like this are of great benefit to the community, but I feel the seasonal and diurnal observations are of great importance to the modeling community. I think these findings could be highlighted more as a useful resource to aerosol modelers and discussed in the conclusion. It would be particularly valuable if the authors could parse out the optical properties at a monthly scale, which could be directly read by a model, for example.

R: Thank you very much for your comments.

We agree with you that the seasonal and diurnal observations are of great importance to the modeling community especially when investigating the aerosol radiative forcing and climate effects. Large bias would be found if the model could not well address the seasonality or diurnal variations. Xu et al. (2016) indicated that the aerosol direct radiative forcing would be underestimated both at the TOA and surface by 2.0 and 38.8 W/m2, respectively, if the diurnal variation were not taken account. Large bias of the aerosol forcing resulting from uncertainties of the aerosol season or diurnal variations would subsequently lead to considerable uncertainties of the climate responses to the aerosol. Analysis on the season and diurnal variations of the aerosol optical properties in this study to some extent are valuable to the modeling-based researches on the aerosol climate effects. Corresponding discussion has been included in Conclusions section of the revised manuscript. Influences of the seasonal and diurnal variations on the aerosol climate effects would be addressed and tested in further study using numerical models.

We also agree with you that the monthly aerosol optical properties are valuable to the climate models. However, much longer observation is needed to ensure the representativeness of the properties in climatology because the monthly values might have substantial inter-annual variability due to serious pollution episodes (suddenly and unpredictable) as presented in Zhuang et al. (2015). We will keep your advice and take this issue into account in the further study based on a much longer measurement. Here, we would like to recommended the readers a research (Che et al., 2015), which

listed the monthly scale columnar aerosol optical depth and angstrom exponent based on a 12-year observations over China.

References:

Xu H., J.P. Guo, X. Ceamanos, Roujean J.L., M. Min, D. Carrer: On the influence of the diurnal variations of aerosol content to estimate direct aerosol radiative forcing using MODIS data, Atmospheric Environment, 141, 186–196. doi:10.1016/j.atmosenv.2016.06.067. 2016.

Zhuang, B. L., Wang, T. J., Liu, J., Ma, Y., Yin, C. Q., Li, S., Xie, M., Han, Y., Zhu, J. L., Yang, X. Q., and Fu, C. B.: Absorption coefficient of urban aerosol in Nanjing, west Yangtze River Delta, China, Atmos. Chem. Phys., 15, 13633–13646, 2015.

Che, H. Z., Zhang, X. Y., Xia, X., Goloub, P., Holben, B., Zhao, H., Wang, Y., Zhang, X. C., Wang, H., Blarel, L., Damiri, B., Zhang, R., Deng, X., Ma, Y., Wang, T., Geng, F., Qi, B., Zhu, J., Yu, J., Chen, Q., and Shi, G..: Ground-based aerosol climatology of China: aerosol optical depths from the China Aerosol Remote Sensing Network (CARSNET) 2002–2013, Atmos. Chem. Phys., 15, 7619–7652, 2015.

Minor Comments:

1. Lines 81 – 102. It would be easier to interpret and compare values presented in this paragraph as a table.

R: Thank you for your suggestion. This part has been rephrased in revised manuscript.

2. Line 29. What is Bsp?

R: It's the aerosol back scattering coefficient. Necessary statement has been included in revised manuscript.

3. Line 30. What is AE?

R: It's the aerosol Ångström exponent. Necessary statement has been included in

revised manuscript.

4. Line 35. What "could be further deteriorated"?

R: It's atmospheric visibility. The sentence has been rephrased to make it more clear in revised manuscript.

5. Lines 38 – 39. Please provide a reference.

R: Two references have been provided in revised manuscript.

6. Line 42. This sentence requires several more references.

R: Several more references have been included in revised manuscript.

7. Lines 59 – 60. How can the bias in Zhuang et al., 2013a be explained by Holler at el al., 2003? One study was 10 years prior to the other.

R: Thank you for your question. Reference Holler et al. (2003) been cited here is not to explain bias in Zhuang et al., 2013a but mainly point out the possible reasons to result in uncertainties of the aerosol radiative forcing. To avoid ambiguity, statements on Holler et al. (2003) have been move to line 54 (line 56 in revised manuscript) after the sentence " Forster et al. (2007) summarized ...".

8. Lines 69 – 70. Please provide a reference.

R: Two references have been provided in revised manuscript.

9. Lines 133. What does ATN stand for?

R: It's light attenuations. Necessary statement has been included in revised manuscript.

Technical Comments:

1. Line 56. Radiative.

R: The word has been corrected.

Relevant changes in revised manuscript:

Author affiliations: 1 and 3 are merged together. The last one is changed to "Department of Climate and Space Sciences and Engineering, University of Michigan, Ann Arbor, Michigan, USA"

In third paragraph of Introduction: Introductions on the aerosol optical properties in lines 70-91 in original manuscript are rephrased and shortened to avoid duplication and confusing, based on the reviewers' comments and suggestions.

In first paragraph of Section 3.2: To make the text more readable, frequency values are listed in a new table (Table 3 in revised manuscript) and corresponding statement has been rephrased according to reviewers' comments.

In fourth paragraph of Section 3.4: Add more discussion on the effect of RH on the aerosol optical properties.

In third and fourth paragraph in Conclusion: According to reviewers' comments, discussions on the importance of the aerosol optical properties seasonal and diurnal variations have been added. Additionally, frequency analysis was shortened.

In Acknowledgements: The foundation number was changed in revised manuscript.

References: References listed and recommended in reviewers' comments were cited and listed in References section in revised manuscript.

In Figure captions: Fig. 8's caption was changed according to reviewers' comments.

Tables: Add a new table (Table 3) to summary the frequencies of the aerosol optical properties. Add a table caption in Table captions section. Table 3 in original manuscript is changed to Table 4.

Figures: Redraw Fig. 9b based on reviewers' comments.

Others: Correct the grammar, cite more references and re-organize some sentences

throughout the manuscript according to reviewers' comments.

Please also note the supplement to this comment:
http://www.atmos-chem-phys-discuss.net/acp-2016-420/acp-2016-420-AC3-supplement.pdf

---

## Author Response (AR2)

**To reviewers:**

Dear reviewers, thank you very much for reviewing the manuscript and providing us the constructive comments and suggestions. With respect to your comments, necessary revisions of the paper have been made. We will response to your comments carefully point by point; details of the revisions can be referred to the revised version of the manuscript.

**Relevant changes of the revised manuscript are listed in the last page.**

**Reviewer #1:**

**Anonymous Referee #1**

I found the revised manuscript to adequately address most of reviewers' concerns. One point I like to add to authors' response to the major comment #3 of the reviewer #1. Authors argue that RH, rather than emissions and other factors, is the most important factor that leads to low SSA in winter and high SSA in summer. However, according to recent studies by Sun et al. [2015], the aerosol composition (as led by emissions) has strong seasonal variability. They found concentration of organic carbonaceous aerosols in Beijing is much higher in winter than other seasons due to enhanced coal combustion in winter. Those high level organic aerosol could also corresponding to a high level black carbon aerosol in winter, which is a strong absorbing aerosol and can lead to low SSA values. So the effect of RH on the seasonal variability of aerosol optical properties (including SSA, Asp, ) is somewhat overstated in the present manuscript. There still needs further leverage and discussion about other influencing factors for aerosol optical properties.

Reference:

Sun, Y. L., Wang, Z. F., Du, W., Zhang, Q., Wang, Q. Q., Fu, P. Q., Pan, X. L., Li, J., Jayne, J., and Worsnop, D. R.: Long-term real-time measurements of aerosol particle composition in Beijing, China: seasonal variations, meteorological effects, and source analysis, Atmos. Chem. Phys., 15, 10149-10165, doi:10.5194/acp-15-10149-2015, 2015.

**R:** Thank you very much for your comments and suggestions. We agree with you that temporal variations of aerosols and trace gases emissions play important roles in the seasonality of the aerosol optical properties, including AAC, SC, EC, SSA, and so on. As discussed in the manuscript, high level of emissions in winter to a great degree directly leads to the largest AAC, Bsp, SC and EC in this season.

**Author response to major comment #3 of the reviewer #1** is mainly to explain the considerable influence of high RH on the aerosol scattering coefficient (rather than SSA) **in summer**. Because large PBL height and precipitation, as well as the lowest air pollutant emissions in summer are all in favor of a smaller SC (similar to AAC or Bsp) in this season. However, the actual SC in summer is substantially larger than that in autumn, in which pollutant emissions are higher. The high RH in summer was thought to be the major reason leading to larger SC. Therefore, the response to major comment #3 of the reviewer #1 was not a statement to emphasize that RH was the most important factor leading to low SSA in winter and high SSA in summer. And the authors do not deny the importance of the emissions either.

Discussions on the variations of SC, AAE and SAE have been included in last version of the manuscript. To avoid overstating the importance of RH on SSA, more discussions on the seasonality of SSA (low in winter and high in summer) have been added to the $3^{rd}$ paragraph of Section 3.1 in the current version of revised manuscript. These statements also figure out the importance of emissions on

the variation of SSA. ASP and RH are highly correlated with each other, which could also be reflected in Fig. 2f, Fig. 5c, Fig. 5d and Fig. 9a, implying that RH might have considerable influence on the aerosol forward scattering coefficient hence SC. Corresponding statement has been added to the revised manuscript ($4^{th}$ paragraph of Section 3.4).

Reference recommended here has been cited in revised manuscript.

I also made out of some editorial corrections listed below.

12: useful to reducing -> useful for reducing

**R:** Corrected.

21: followed -> follow

**R:** Corrected.

21-22: please rephrase the following sentence in a clearer way: "the ranges around their averages …"

**R:** The sentence has been rephrased in revised manuscript to make it much clearer.

63: the observations -> observations

**R:** Corrected.

63: the observed -> observed

**R:** Corrected.

65: among countries in East Asia and even the world -> worldwide.

**R:** Corrected.

66: aerosols -> anthropogenic aerosols

**R:** Corrected.

89: substantial observation-based studies mentioned -> intensive observation-based studies

**R:** Corrected.

99: Conclusions -> conclusions

**R:** Corrected.

108: wavelength dependent -> wavelength-dependent (and elsewhere of the manuscript)

**R:** Corrected throughout the manuscript.

160: as followed -> by

**R:** Corrected.

176: It's -> It is

**R:** Corrected.

177: single wavelength -> single-wavelength, or monochromatic

**R:** Corrected.

184: back scattering -> backscattering

**R:** Corrected.

199: winter times -> winter

**R:** Corrected.

200: temporal trend -> temporal variability

**R:** Corrected.

200: SSA's -> that of SSA

**R:** Corrected.

202: the other -> other

**R:** Corrected.

370: is more -> are more

**R:** Corrected.

400: long distance -> long-distance

**R:** Corrected.

**Relevant changes in revised manuscript:**

**Result and discussions:** Add discussions of aerosol optical properties in 3$^{rd}$ paragraph of Section 3.1 and 4$^{th}$ paragraph of Section 3.4.

**In Acknowledgements:** Add one more foundation number. And thank the reviewers for their constructive and valuable comments.

**References:** References listed and recommended in reviewers' comments were cited in the text and listed in References section in revised manuscript.

**Corrections:** Grammar and editorial corrections are made throughout the manuscript according to reviewer's suggestion.